# Muscular TOR knockdown and endurance exercise ameliorate high salt and age-related skeletal muscle degradation by activating the MTOR-mediated pathway

**Shi-jie Wang**, **Deng-tai Wen**\*, **Ying-hui Gao**, **Jing-feng Wang**, **Xing-feng Ma**

Physical Culture Institute Ludong University, City Yantai, Shandong Province, China

\* dt.wen@foxmail.com

## Abstract

The target of rapamycin(TOR)gene is closely related to metabolism and cellular aging, but it is unclear whether the TOR pathways mediate endurance exercise against the accelerated aging of skeletal muscle induced by high salt intake. In this study, muscular TOR gene overexpression and RNAi were constructed by constructing Mhc$^{GAL4}$/TOR-overexpression and Mhc$^{GAL4}$/TOR$^{UAS-RNAi}$ systems in Drosophila. The results showed that muscle TOR knockdown and endurance exercise significantly increased the climbing speed, climbing endurance, the expression of autophagy related gene 2(ATG2), silent information regulator 2 (SIR2), and ppary coactivator 1(PGC-1α) genes, and superoxide dismutases(SOD) activity, but it decreased the expression of the TOR gene and reactive oxygen species(ROS) level, and it protected the myofibrillar fibers and mitochondria of skeletal muscle in Drosophila on a high-salt diet. TOR overexpression yielded similar results to the high salt diet(HSD) alone, with the opposite effect of TOR knockout found in regard to endurance exercise and HSD-induced age-related skeletal muscle degradation. Therefore, the current findings confirm that the muscle TOR gene plays an important role in endurance exercise against HSD-induced age-related skeletal muscle degeneration, as it determines the activity of the mammalian target of rapamycin(MTOR)/SIR2/PGC-1α and MTOR/ATG2/PGC-1α pathways in skeletal muscle.

## 1 Introduction

Preserving or restoring muscle mass is essential for healthy aging, and studies have shown that aging is associated with muscle resistance to anabolic effects, a progressive loss of skeletal muscle mass and function in older adults, often referred to as "sarcopenia" [1, 2]. Recent studies have shown that excessive salt intake may be associated with fat accumulation and muscle weakness, and these phenomena may increase the risk of sarcopenia [3], we speculate that reducing salt intake may have a preventive effect on sarcopenia. In addition, a high-salt diet can lead to multiple chronic comorbidities, including hypertension, heart failure, and

**Data Availability Statement:** All relevant data are within the manuscript and its Supporting Information files.

**Funding:** This research was supported by the National Natural Science Foundation of China (No. 32000832), the Shandong provincial college youth innovation and technology support program (2023RW05). The funders had no role in study design, data collection and analysis, decision to publish, or preparation of the manuscript.

**Competing interests:** The authors have declared that no competing interests exist.

increased mortality. Studies on aging Drosophila have shown that HSD negatively affects age-related climbing ability and mortality by up-regulating salt gene expression and inhibiting the drosophila forkhead box O(dFOXO)/SOD pathway, resulting in impaired climbing ability and longevity in aging Drosophila [4, 5]. Studies on Drosophila have shown that flight exercise can effectively improve the preference and adverse effects of wild fruit flies on HSD, and regular aerobic exercise can inhibit the adverse effects of HSD on mitochondrial function, physiology, and feeding behavior of Drosophila [5]. However, in the aging process, the molecular mechanism of exercise against skeletal muscle aging caused by a high-salt diet remains to be clarified.

TOR plays a central role in related energy metabolism and aging processes, as well as age-related diseases [6]. Recent studies have shown that TOR is closely related to cellular senescence [7]. For example, TOR gene knockdown in fruit flies has a very positive effect on promoting autophagy and delaying skeletal muscle aging and longevity [8]. Inhibition of TORC1 activity leads to decreased protein synthesis and increased autophagy [9]. A decrease in anabolism reduces the production of ROS in cells, while an increase in autophagy and glycolysis increases the cells' resistance to ROS [10]. Up-regulation of autophagy by rapamycin (a TOR inhibitor) reduces ATG2 and PGC-1α deficit-induced aging [5]. In addition to the autophagy regulation of aging, TOR can also participate in the regulation of aging independently [11]. In addition, TOR inhibition was associated with significant weight loss, and studies have shown that rapamycin can reduce body weight and lipid levels in high-fat-fed mice [12]. Interestingly, a high-salt diet induces ROS production, which leads to oxidative damage [13]. Exercise can enhance the ability of skeletal muscle to resist oxidative stress and reduce oxidative damage [14]. This seems to indicate that the skeletal muscle TOR gene is involved in the regulation of skeletal muscle aging and may also mediate the effects of exercise and high salt intake on skeletal muscle aging.

In this study, we constructed the mhc$^{UAS-Gal4}$/TOR$^{OE}$ (experimental containing the UAS-OE gene and the Mhc-GAL4 gene) system and the mhc$^{Gal4}$/TOR$^{UAS-RNAi}$ system to construct TOR gene overexpression or knockdown in Drosophila skeletal muscle. Then, Drosophila skeletal muscle function (climbing index, climbing to fatigue time), SOD activity, ROS level, muscle MTOR gene, ATG2 gene, PGC-1α gene, and SIR2 gene expression were detected to determine whether the skeletal muscle TOR gene was involved in regulating exercise against muscle aging caused by Drosophila HSD(the research process is shown in Fig 1).

## 2 Material and method

### 2.1 Drosophila strains and normal dietary configuration

TOR$^{UAS-OE}$ (control lacking the Mhc-GAL4 gene) Fruit Fly (stock ID: 7013; FlyBase genotype: y1 w*; P{UAS-TOR.TED}II) and MHC$^{gal4}$ (stock ID: 55,133; FlyBase genotype: w *; P{Mhc-GAL4.K} 2/TM3, Sb 1) Drosophila were obtained from the Bloomington Inventory Center. TOR$^{UAS-RNAi}$ Drosophila (Stock ID: v318201; FlyBase genotype: PBac{fTRG00713. sfGFP-TVPTBF}VK00033 obtained from the Drosophila Resource Centre, Vienna.

Common food contains soybean flour(1.6%), yeast(2.0%), corn flour(6.7%), AGAR(0.7%), sucrose(4.8%), maltose(4.8%), propionic acid(0.3%). High-salt foods can be made by adding 2% sodium chloride to common foods [4]. Taking 1 L as an example, for ordinary meals, 42 g of corn flour, 10 g of soybean flour, 13 g of yeast powder and 8g of AGAR strips are added to the pot, and 1 liter of pure water is added. Stir continuously during the heating process to completely melt the AGAR strips until the solution boils. After boiling, stop heating, and add 31g of sucrose and maltose each during cooling. After sucrose and maltose are completely dissolved, add 2000 uL propionate and 1 g sodium phenylpropionate as preservatives, stir thoroughly, and immediately pack into a clean culture tube with a thickness of about 0.5 cm each.

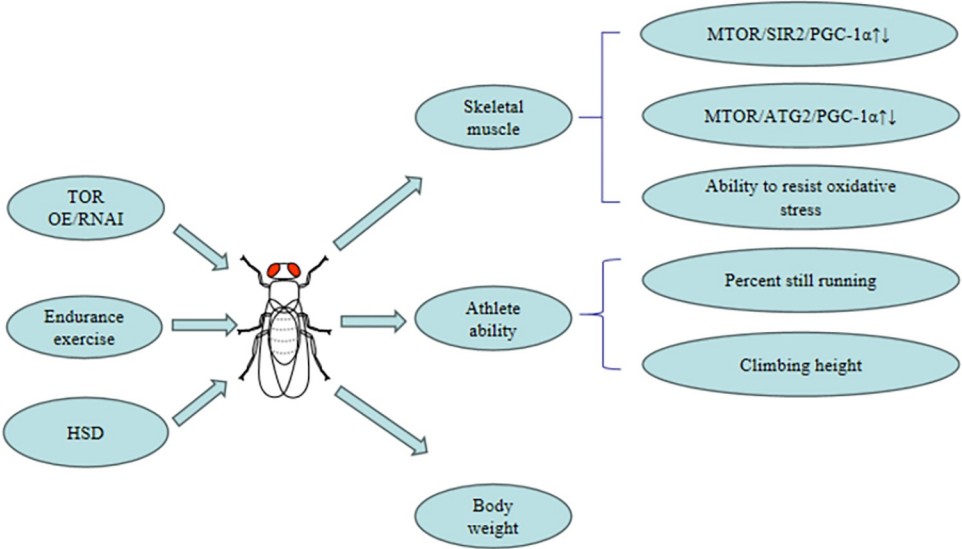

**Fig 1. Muscle TOR gene, endurance exercise activates the MTOR/ SIR2/PGC-1α and MTOR/ATG2/PGC-1α pathway and regulates age-related decline in skeletal muscle.**

Other sizes of fruit fly media are converted according to this ratio. During the experiment, all experimental fruit flies were housed in a 12-h light/dark cycle and a 25° C incubator with 50% humidity. All experimental fruit flies were fed fresh food every other day during the experiment [15].

## 2.2 Hybridization scheme and grouping

The $TOR^{UAS}/mhc^{Gal4}$ system was constructed by hybridization to achieve the regulation of TOR gene differential expression. Male $TOR^{UAS-OE}$ flies were hybridized with female $mhc^{Gal4}$ flies, and F1 generation $TOR^{UAS-OE} >$ mhcGal4 male flies were collected. They were divided into the TOR overexpression group ($TOR^{OE}$), the TOR overexpression exercise group ($TOR^{OE} + E$), the TOR overexpression high salt group ($TOR^{OE} + HSD$), and the TOR overexpression high salt exercise group ($TOR^{OE} + HSD + E$). Male $TOR^{UAS-RNAi}$ flies were hybridized with female $mhc^{Gal4}$ flies, and F1-generation $TOR^{UAS-RNAi} > mhc^{Gal4}$ male flies were collected. They were divided into TOR knockdown group ($TOR^{RNAi}$), the TOR knock-down exercise group ($TOR^{RNAi} + E$), the TOR knock-down high-salt exercise group ($TOR^{RNAi} + HSD$) and the TOR knock-down high-salt exercise group ($TOR^{RNAi} + HSD + E$). In order to eliminate the influence of genetic background on the experiment, F1 generation male flies with $TOR^{UAS-OE}$ and $TOR^{UAS-RNAi}$ were used as a genetic control group. They were: TOR normal expression group (overexpressed gene control group $TOR^{UAS-OE}$, knockdown gene control group $TOR^{UAS-RNAi}$), TOR normal expression exercise group (overexpressed gene control exercise group $TOR^{UAS-OE} + E$, knockout gene control exercise group $TOR^{UAS-RNAi} + E$), TOR normal expression high salt group (overexpressed base $TOR^{UAS-OE} + HSD$ in the control high-salt group, $TOR^{UAS-RNAi} + HSD$ in the knockout gene control high-salt group, and TOR normal expression in the high-salt exercise group ($TOR^{UAS-OE} + HSD + E$ in the overexpressed gene control high-salt exercise group and $TOR^{UAS-RNAi} + HSD + E$ in the knockout gene control high-salt exercise group). There are 16 groups of 400 animals in each group, $TOR^{UAS-OE}$, $TOR^{UAS-OE} + E$, $TOR^{UAS-OE} + HSD$, $TOR^{UAS-OE} + HSD + E$, $TOR^{OE} + E$, $TOR^{OE} + HSD$, $TOR^{OE} + HSD + E$, $TOR^{UAS-RNAi}$, $TOR^{UAS-RNAi} + E$, $TOR^{UAS-RNAi} + HSD$, $TOR^{UAS-RNAi} + HSD + E$, $TOR^{RNAi}$, $TOR^{RNAi} + E$, $TOR^{RNA\ i} + HSD$, $TOR^{RNAi} + HSD + E$.

## 2.3 Exercise training

According to the anti-gravity climbing characteristics of fruit fly [16], fruit fly endurance training can be applied to fruit fly using a fruit fly endurance training exercise device. The principle is that when the fruit fly at the bottom of the test tube climbs to the top of the test tube, the test tube starts to rotate at a constant speed of 60 rad/s until the top and bottom exchange, at which time the fruit fly climbs to the top again, and so on [17]. An area of 8 cm was reserved from the bottom of the culture tube to the bottom of the cotton plug for Drosophila movement, and there was 10 second of climbing time after the two ends of the test tube switched directions. All exercise groups started at three weeks and continued exercise training for two weeks, each exercise to exhaustion, one day off for every two training days, and five weeks. The exercise type belonged to aerobic endurance exercise. All high-salt fruit flies were fed a high-salt diet starting at five weeks and were put on a high-salt diet plan for two weeks(refer to WEN et al. 's study) [18].

Climbing endurance measurement: 15 fruit flies were randomly selected from each group and placed into 15 test tubes (including media), and the test tubes were rotated by the fruit fly exercise trainer. Using the fruit fly's climbing properties, watch the fruit fly climb to the top of the test tube. As the flies cycled up and eventually failed to reach the top, the time it took the flies to exhaust themselves was recorded.

Climb rate measurement: 120 fruit flies were randomly selected from each group, 30 fruit flies per tube. Each tube of fruit flies is placed individually under a high-definition camera and prepared for recording. After turning on the camera, shake the fruit flies in the tube to the bottom of the tube every 15 seconds, and after five shocks, replace the flies with the next tube. The best climb times were selected from 5 climbs per tube for data processing.

## 2.4 Transmission electron microscopy of skeletal muscle

Skeletal muscle was dissected in a cold fixative (Servicebio, G1102, Wuhan, Hubei, China; 2.5% glutaraldehyde in 0.1 M PIPES buffer, pH 7.4) according to electron microscopy analysis. After fixation at 4°C for 10 hours, the sample was cleaned with 0.1 M PIPES, fixed in 1% OsO4 (30 minutes), and stained in 2% uranedioxyacetate (1 hour). Samples are dehydrated in an ethanol series (Sinopharm Group Chemical Reagent Co., LTD, 100092183, Shanghai, China; 50%, 70%, 100%) and embedded in epoxy resin(SPI,90529-77-4, Beijing, China). Ultra-thin sections. (50 nm) were cut and observed on a Tecnai G2 Spirit Bio-TWIN electron microscope (HITACHI, HT7800/HT7700, Beijing, China).

## 2.5 Myosin heavy chain detection

The chest muscles of five-week-old Drosophila were taken. Remove the frozen sections from the -20°C refrigerator and restore them to room temperature. Fix them with tissue fixing solution (Servicebio, G1101) for 15 minutes, wash them with tap water, and let them dry. Part of the saturated oil red O dye solution (Servicebio, G1015) and 4 parts of distilled water were fully mixed and evenly, placed in a water bath at 60–70°C for 30 min, and then filtered with qualitative filter paper. After natural cooling, the oil red O working solution was obtained. Slice into the oil red dye and soak for 8–10 minutes(cover and avoid light). Remove the slices, stay for 3 s, and then immerse in two cylinders of 60% isopropyl alcohol (Sinopsin Chemical Reagent Co., LTD., 80109218) successively for 3 s and 5 s. Take out the slices, stay for 3 s, dip in hematoxylin (Servicebio, G1004), dye for 3–5 min, and soak in 3 tanks of pure water for 5 s, 10 s, and 30 s. The differentiation solution (Servicebio, G1039) was differentiated for 2–8 s, and the two tanks of distilled water were washed for 10 s each, and the return blue solution (Servicebio, G1040) was returned to the blue for 1 s. The slices were gently immersed in the

two tanks of tap water for 5 s and 10 s each, and the staining effect was examined by microscopy. Glycerin gelatin sealing tablets (Servicebio, G1402), microscopy, image acquisition, and analysis.

## 2.6 Enzyme linked immunosorbent assay

SOD activity levels and ROS levels were determined by ELISA (Insect SOD and ROS ELISA kit, Fankew, Shanghai Kexing Trading Co., Ltd). The chest muscles of 30 fruit flies were homogenized in PBS (Servicebio, G0002, pH 7.2–7.4). The sample is quickly frozen in liquid nitrogen and then kept at 2–8˚C after melting. The samples were homogenized with a grinder (Servicebio, KZ-Ⅲ-FP) and centrifuged at 2000–3000 rpm for 20 min. Then, we remove the supernatant. Determination: Zero the blank hole and read the absorbance at 450 nm within 15 minutes after adding the termination liquid(Servicebio, G0027).

## 2.7 Real-time quantitative PCR

About 20 flies' chest muscles were homogenized in Trizol. First, 10 µg of total RNA was extracted and purified from Trizol (Servicebio, G3013; Trizol, Invitrogen) with organic solvent. Purified RNA was treated with DNase I (Servicebio, G3342; RNase-free, Roche) and used to produce an oligo-DT primer cDNA (SuperScript II RT, Invitrogen), which was then used as a template for quantitative real-time PCR. The rp49 gene was used as an internal reference to standardize the total RNA count. Real-time PCR was performed with SYBR green using the ABI7300 real-time PCR instrument (Servicebio, GM2007; Applied Biosystems), and 3 biological replicates were performed. The expression of various genes was determined by comparing CT methods (ABI Prism 7700 Sequence Detection System User Bulletin #2, Applied Biosystems). The primer sequence for TOR is as follows: S (sense primer) 5′-TACAGCAGGCACT CCTTCAAATA-3′; A (antisense primer): 5′-ACCGTGCGATCCTTCTCCTT-3′. The primer sequence of ATG2 is as follows: S: 5′-GTTTGGACCCACTCCCTACGA-3′; A: 5′-AACTT CTTGAATCC GGTGGG-3′.The sequence of PGC-1α primers is as follows: S: 5′-ACCT GGCGATTCTGATTATGACT-3′; A: 5′-CCTTTACATTGTCCACATAGCGT-3′.The SIR2 primer sequence is as follows: S: 5′-GTTTCCAGGAGCCCAGGTATT-3′; A:5′-CAGTGAA GGCGGTAGCAATG-3′. Primer sequence of the internal reference: Rp49-F: 5′-CTAAGCTG TCGCACAAATGG-3′; R: 5′-AACTTCTTGAATCCGGTGGG-3′.

## 2.8 Statistic analysis

Independent sample tests were used to assess the differences between 1-week-old flies and 5-week-old flies. One-way analysis of variance (ANOVA) and minimum significant difference (LSD) tests were used to identify differences between these groups. The P-values of life curve and climbing endurance curve were calculated by the logarithmic rank test. Analysis was performed using the Social Science Statistical Package (SPSS) version 16.0 for Windows (SPSS Inc., Chicago, USA), with statistical significance set to $P < 0.05$. Data are expressed as mean ± SEM.

# 3 Results

## 3.1 Comparison of MTOR gene expression levels in overexpressed/knock-down drosophila aged 3 weeks

In order to verify the successful construction of the overexpression/knockdown system in Drosophila melanogaster, and to rule out the influence of age/exercise ability measurement on the expression level of MTOR gene in drosophila melanogaster, the expression level of MTOR

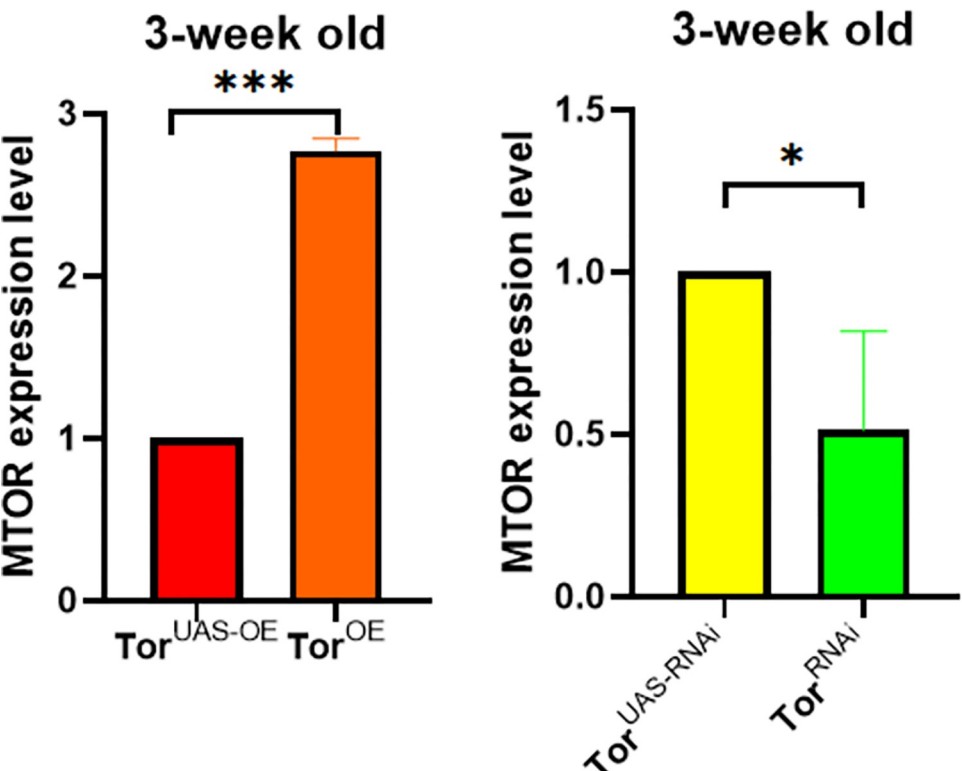

**Fig 2. Comparison of MTOR gene expression levels in overexpressed/knock-down Drosophila aged 3 weeks.** Data were expressed by mean ±SEM. * P < 0.05; * * P < 0.01; * * * P < 0.001; n indicates no significant difference. The sample size was muscle of 20 flies per group, measured 3 times.

gene in drosophila melanogaster was compared in pairs between the overexpression group, the control group and the knockdown group at the age of three weeks. It was found that the expression level of MTOR gene in TOR$^{UAS-OE}$ group and TOR$^{OE}$ group was significantly different (P<0.01), which was higher in TOR$^{OE}$ group than in TOR$^{UAS-OE}$ group. There were also significant differences in the expression level of MTOR gene between TOR$^{UAS-RNAi}$ group and TOR$^{R-NAi}$ group (P<0.05), specifically, the TOR$^{RNAi}$ group was lower than the TOR$^{UAS-RNAi}$ group (Fig 2).

### 3.2 Exercise prevents age-related degeneration of climbing ability caused by HSD in the muscles of Drosophila that normally express TOR

In muscular TOR$^{UAS-OE}$ flies, results showed that both exercise and HSD reduced body weight (P < 0.05 or P < 0.01) (Figs 3E and 4E). There was no significant difference in the climbing height within 3 seconds of flies without intervention at weeks (P > 0.05), and exercise had no significant effect on the climbing height within 3 seconds of flies aged 4 weeks (P > 0.05), but significantly increased the climbing height within 3 seconds of flies aged 5 weeks (P < 0.001) (Fig 3F). HSD significantly reduced the 3-second climbing height of 4-week-old fruit flies (P < 0.05), but had no significant effect on the 3-second climbing height of 5-week-old fruit flies (P > 0.05) (Fig 4F), and exercise significantly changed the 3-second climbing height reduction of 4.5 week-old fruit flies induced by HSD (P < 0.05 or P < 0.001) (Fig 5F). There was no significant difference in the climbing to fatigue time (CTF) of 1.3 week-aged flies in muscular TOR$^{UAS-OE}$ flies (P > 0.05) (Fig 3A and 3B), exercise significantly increased CTF of

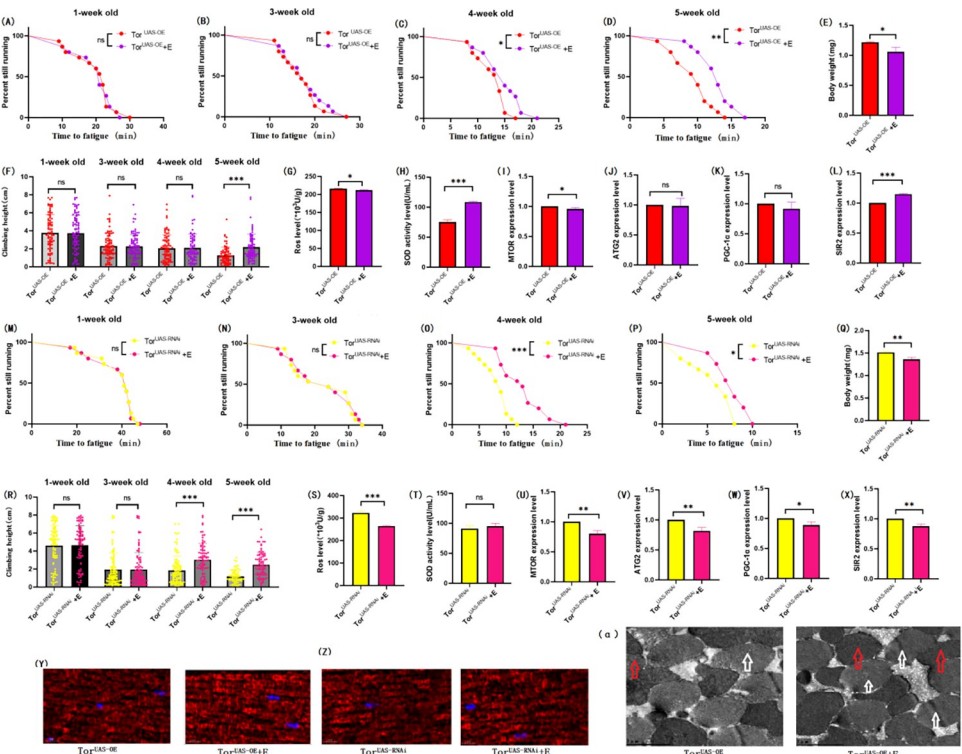

**Fig 3. Effects of exercise on body weight, climbing ability and skeletal muscle physiological structure of TOR$^{UAS-OE}$ and TOR$^{UAS-RNAi}$ flies.** (A) Fatigue time of 1-week-old flies in TOR$^{UAS-OE}$ group. (B) Fatigue time of 3-week-old flies in TOR$^{UAS-OE}$ group. (C) Fatigue time of flies aged 4 weeks in TOR$^{UAS-OE}$ group. (D) Fatigue time of 5-week-old flies in TOR$^{UAS-OE}$ group. (E) Body weight of 5-week-old flies in TOR$^{UAS-OE}$ group. (F) Climb height of 1,3,4,5 week-old flies in TOR$^{UAS-OE}$ group within 3 seconds. (G) Skeletal muscle ROS levels in TOR$^{UAS-OE}$ group. (H) SOD level in skeletal muscle of TOR$^{UAS-OE}$ group. (I) Relative expression of TOR gene in skeletal muscle of TOR$^{UAS-OE}$ group. (J) Relative expression of ATG2 gene in skeletal muscle of TOR$^{UAS-OE}$ group. (K) Relative expression of PGC-1α gene in skeletal muscle of TOR$^{UAS-OE}$ group. (L) Relative expression of SIR2 gene in skeletal muscle of TOR$^{UAS-OE}$ group. (M) Fatigue time of 1-week-old flies in TOR$^{UAS-RNAi}$ group. (N) Fatigue time of 3-week-old flies in TOR$^{UAS-RNAi}$ group. (O) Fatigue time of flies aged 4 weeks in TOR$^{UAS-RNAi}$ group. (P) Fatigue time of 5-week-old flies in TOR$^{UAS-RNAi}$ group. (Q) Body weight of 5-week-old flies in TOR$^{UAS-RNAi}$ group. (R) Climb height of 1,3,4,5 week-old flies in TOR$^{UAS-RNAi}$ group within 3 seconds. (S) Skeletal muscle ROS levels in TOR$^{UAS-RNAi}$ group. (T) SOD level in skeletal muscle of TOR$^{UAS-RNAi}$ group. (U) Relative expression of TOR gene in skeletal muscle of TOR$^{UAS-RNAi}$ group. (V) Relative expression of ATG2 gene in skeletal muscle of TOR$^{UAS-RNAi}$ group. (W) Relative expression of PGC-1α gene in skeletal muscle of TOR$^{UAS-RNAi}$ group. (X) Relative expression of SIR2 gene in skeletal muscle of TOR$^{UAS-RNAi}$ group. (Y) myosin heavy chain immunofluorescence in TORUAS-OE group, with blue dots as nucleus and red fluorescent bands as myofibril. HSD reduces fluorescence expression (scale: white line is 5 microns). (Z) TOR$^{UAS-RNAi}$ group myosin heavy chain immunofluorescence. (α) Skeletal muscle transmission electron microscopy (scale: small mesh represents 0.2 microns). The white arrow points to the myofibril. The red arrow points to the mitochondria. Climbing height measurements, the sample size of each group is about 150–170 animals. Univariate analysis of variance (ANOVA) and minimum significance difference (LSD) tests were used to determine differences between groups. The P-values of climbing endurance curve and survival curve were calculated by log-rank test. Data were expressed by mean ±SEM. * P < 0.05; * * P < 0.01; * * * P < 0.001; n indicates no significant difference. The sample size of proteins and ROS was muscle of 20 flies per group, measured 3 times. The sample size for RT-PCR and ELISA was muscle of 20 flies per group, measured 3 times. Univariate analysis of variance (ANOVA) and minimum significance difference (LSD) tests were used to determine differences between groups. (A) Data are represented by mean ±SEM. * P < 0.05; * * P < 0.01; n indicates no significant difference.

4.5 week-aged flies (P < 0.05 or P < 0.01) (Fig 3C and 3D), and HSD significantly decreased CTF of 4.5 week-aged flies (P < 0.01) (Fig 4C and 4D). Exercise significantly improved HSD-induced CTF reduction (P < 0.05 or P < 0.001) (Fig 5C and 5D).

In addition, in muscular TOR$^{UAS-OE}$ Drosophila, exercise significantly decreased the expression of the MTOR gene (P < 0.05) (Fig 3I). It had no significant effects on the

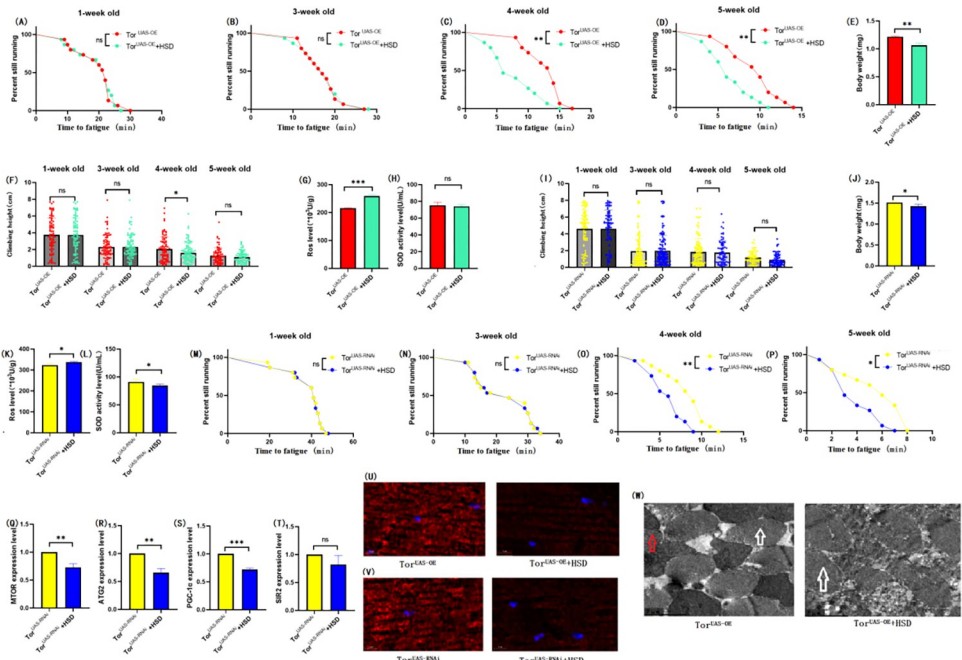

**Fig 4. Effects of HSD on body weight, climbing ability and skeletal muscle physiological structure of TOR^UAS-OE and TOR^UAS-RNAi flies.** (A) Fatigue time of 1-week-old flies in TOR^UAS-OE group. (B) Fatigue time of 3-week-old flies in TOR^UAS-OE group. (C) Fatigue time of flies aged 4 weeks in TOR^UAS-OE group. (D) Fatigue time of 5-week-old flies in TOR^UAS-OE group. (E) Body weight of 5-week-old flies in TOR^UAS-OE group. (F) Climb height of 1,3,4,5 week-old flies in TOR^UAS-OE group within 3 seconds. (G) Skeletal muscle ROS levels in TOR^UAS-OE group. (H) SOD level in skeletal muscle of TOR^UAS-OE group. (I) Climb height of 1,3,4,5 week-old flies in TOR^UAS-RNAi group within 3 seconds. (J) Body weight of 5-week-old flies in TOR^UAS-RNAi group. (K) Skeletal muscle ROS levels in TOR^UAS-RNAi group. (L) SOD level in skeletal muscle of TOR^UAS-RNAi group. (M) Fatigue time of 1-week-old flies in TOR^UAS-RNAi group. (N) Fatigue time of 3-week-old flies in TOR^UAS-RNAi group. (O) Fatigue time of flies aged 4 weeks in TOR^UAS-RNAi group. (P) Fatigue time of 5-week-old flies in TOR^UAS-RNAi group. (Q) Relative expression of TOR gene in skeletal muscle of TOR^UAS-RNAi group. (R) Relative expression of ATG2 gene in skeletal muscle of TOR^UAS-RNAi group. (S) Relative expression of PGC-1α gene in skeletal muscle of TOR^UAS-RNAi group. (T) Relative expression of SIR2 gene in skeletal muscle of TOR^UAS-RNAi group. (U) myosin heavy chain immunofluorescence in TOR^UAS-OE group, with blue dots as nucleus and red fluorescent bands as myofibril. HSD reduces fluorescence expression (scale: white line is 5 microns). (V) TOR^UAS-RNAi group myosin heavy chain immunofluorescence. (W) Skeletal muscle transmission electron microscopy (scale: small mesh represents 0.2 microns). The white arrow points to the myofibril. The red arrow points to the mitochondria. Climbing height measurements, the sample size of each group is about 150–170 animals. Univariate analysis of variance (ANOVA) and minimum significance difference (LSD) tests were used to determine differences between groups. The P-values of climbing endurance curve and survival curve were calculated by log-rank test. Data were expressed by mean ±SEM. * P < 0.05; * * P < 0.01; * * * P < 0.001; n indicates no significant difference. The sample size of proteins and ROS was muscle of 20 flies per group, measured 3 times. The sample size for RT-PCR and ELISA was muscle of 20 flies per group, measured 3 times. Univariate analysis of variance (ANOVA) and minimum significance difference (LSD) tests were used to determine differences between groups. (A) Data are represented by mean ±SEM. * P < 0.05; * * P < 0.01; n indicates no significant difference.

expression of ATG2 (Fig 3J) and PGC-1α genes (Fig 3K; P > 0.05), but significantly increased the expression of the SIR2 gene (Fig 3L; P < 0.001), significantly increased the activity of SOD in muscle (Fig 3H; P < 0.001), and significantly decreased the level of ROS (Fig 3G; P < 0.05). HSD decreased the expression of MTOR(Fig 4Q), ATG2 (Fig 4R), and PGC-1α genes (Fig 4S; P < 0.01 or P < 0.001), had no significant effect on the expression level of the SIR2 gene (Fig 4T; P > 0.05), decreased the activity of SOD (Fig 4L; P < 0.05), and increased the level of ROS (Fig 4K; P < 0.05), exercise significantly improved the changes induced by HSD (Fig 5A–5L).

Myosin heavy chain red immunofluorescence images showed that HSD reduced myosin heavy chain fluorescence expression and muscle fiber alignment (Fig 4U and 4V), which could

be reversed by exercise (Fig 5Y and 5Z). Skeletal muscle transmission electron microscopy images showed that exercise maintained the normal myofibrillar and mitochondrial morphological structures (Fig 3α), but HSD destroyed the myofibrillar and mitochondrial morphological structures (Fig 4W), and exercise improved the damage to skeletal muscle myofibrillar and mitochondrial morphological structures induced by HSD (Fig 5α).

In muscular TOR$^{UAS-RNAi}$(TOR-normal-expression) Drosophila melanogaster, exercise decreased the expression of the MTOR gene (Fig 3U; P < 0.01), ATG2 gene (Fig 3V; P < 0.01), PGC-1α gene (Fig 3W; P < 0.05), and SIR2 gene (Fig 3X; P < 0.01). Exercise combined with HSD decreased the expression of the MTOR gene (Fig 5U; P < 0.05), ATG2 gene (Fig 5V; P < 0.01),PGC-1α gene (Fig 5W; P < 0.001) and SIR2 gene (Fig 5X; P < 0.05). The effect of HSD on gene expression was basically consistent with that of TOR$^{UAS-OE}$ Drosophila. The effects of exercise, HSD, and exercise combined with HSD on climbing height (Figs 3R, 4Q and 5R), climbing endurance (Figs 3M–3P, 4M–4P and 5M-5P), antioxidant capacity (Figs 3T, 3Z, 4L, 4V, 5T and 5Z), ROS level myofibrillar fiber, and mitochondria were basically consistent with those of TOR$^{UAS-OE}$ Drosophila (Figs 3S, 4K and 5S).

## 3.3 Muscle TOR overexpression inhibition exercise against HSD-induced skeletal muscle aging

To further investigate the role of muscle TOR/ATG2/PGC-1α and TOR/SIR2/PGC-1α pathways in exercise against HSD-induced age-related skeletal muscle degradation, we constructed the mhc-Gal4/TOR-UAS overexpression (TOR$^{OE}$) system. Overexpression of the TOR gene in Drosophila skeletal muscle. The results showed that muscle TOR gene overexpression significantly increased the body weight of TOR$^{OE}$ flies (Fig 6E; P < 0.001), and both exercise and HSD significantly decreased the body weight of TOR$^{OE}$ flies (Figs 7E, 8 E; P < 0.01).Muscle TOR$^{OE}$ significantly decreased the climbing height of 1, 3, 4, and 5 week-old flies (Fig 6F; P < 0.05 or P < 0.01 or P < 0.001), and exercise significantly increased the climbing height of 4, 5 week-old TOR$^{OE}$ and TOR$^{OE}$ + HSD flies (Figs 7F, 8F; P < 0.05 or P < 0.001). HSD did not significantly reduce the climbing height of TOR$^{OE}$ flies (Fig 8F; P > 0.05).

The results showed that TOR$^{OE}$ significantly decreased the CTF of flies at the end of 1, 3, 4, and 5 weeks of age (Fig 6A–6D; P < 0.05 or P < 0.001). Exercise significantly increased CTF in TOR$^{OE}$ and TOR$^{OE}$ + HSD flies (Figs 7A–7D and 9A–9D; P < 0.05 or P < 0.01 or P < 0.001). HSD significantly decreased CTF in TOR$^{OE}$ flies (Fig 8A–8D; P < 0.05 or P < 0.001).

Muscle TOR$^{OE}$ significantly increased the expression levels of muscle MTOR gene (Fig 6I; P < 0.001), ATG2 gene (Fig 6J; P < 0.001), SIR2 gene (Fig 6L), and PGC-1α gene (Fig 6K; P < 0.001), and significantly increased muscle ROS level (Fig 6G; P < 0.001). However, SOD activity in muscle was significantly decreased (Fig 6H; P < 0.05). Myosin heavy chain red immunofluorescence images showed that TOR$^{OE}$ promoted the decrease of HSD-induced myosin heavy chain fluorescence expression and muscle fiber arrangement (Fig 7M).Transmission electron microscopy (TEM) images of skeletal muscle showed that TOR$^{OE}$ damaged the morphological structure of myofibril and mitochondria (Fig 6N).

In TOR$^{OE}$ and TOR$^{OE}$ + HSD Drosophila, exercise significantly decreased the expressions of MTOR gene (Fig 9I; P < 0.001), ATG2 gene (Fig 9J; P < 0.001), PGC-1α gene(Fig 9K), and SIR2 gene (Fig 8L; P < 0.001), and significantly increased the activity of SOD in muscle (Fig 8 H; P < 0.05). However, muscle ROS levels were significantly decreased (Fig 9G; P < 0.001). Myosin heavy chain red immunofluorescence images showed that exercise prevented the HSD-induced reduction of myosin heavy chain fluorescence expression and muscle fiber arrangement (Fig 9M).

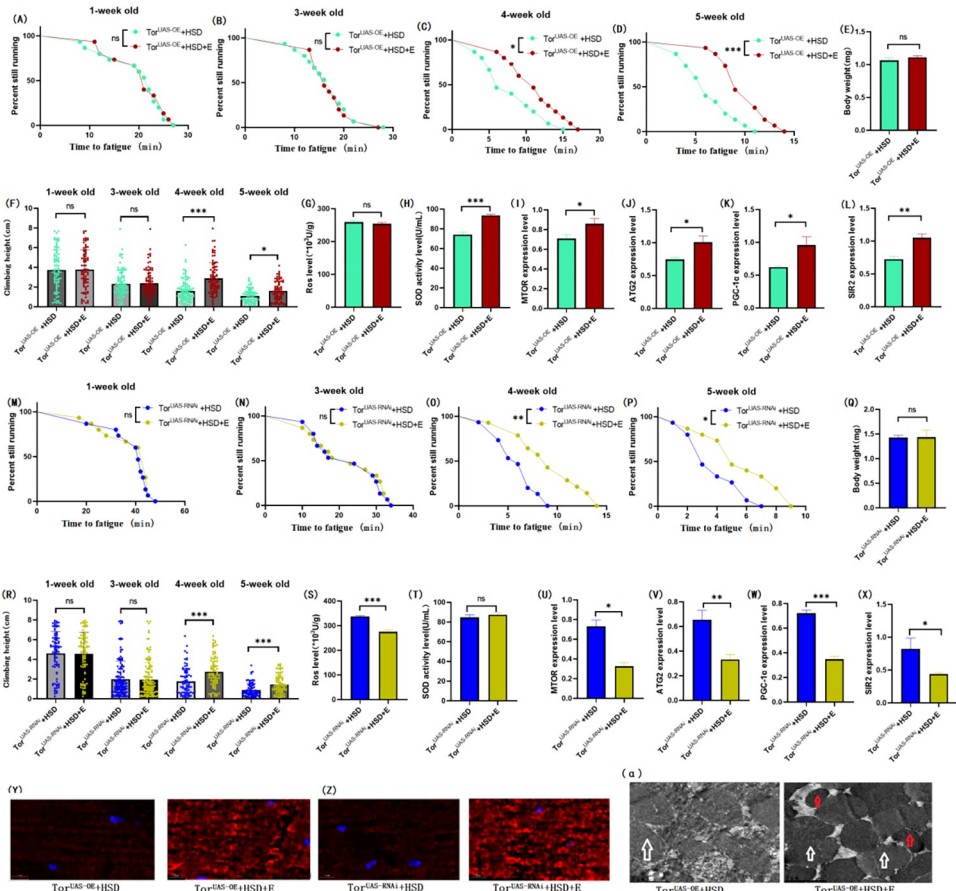

**Fig 5. Effects of exercise and HSD on body weight, climbing ability and skeletal muscle physiological structure of TOR^UAS-OE and TOR^UAS-RNAi flies.** (A) Fatigue time of 1-week-old flies in TOR^UAS-OE group. (B) Fatigue time of 3-week-old flies in TOR^UAS-OE group. (C) Fatigue time of flies aged 4 weeks in TOR^UAS-OE group. (D) Fatigue time of 5-week-old flies in TOR^UAS-OE group. (E) Body weight of 5-week-old flies in TOR^UAS-OE group. (F) Climb height of 1,3,4,5 week-old flies in TOR^UAS-OE group within 3 seconds. (G) Skeletal muscle ROS levels in TOR^UAS-OE group. (H) SOD level in skeletal muscle of TOR^UAS-OE group. (I) Relative expression of TOR gene in skeletal muscle of TOR^UAS-OE group. (J) Relative expression of ATG2 gene in skeletal muscle of TOR^UAS-OE group. (K) Relative expression of PGC-1α gene in skeletal muscle of TOR^UAS-OE group. (L) Relative expression of SIR2 gene in skeletal muscle of TOR^UAS-OE group. (M) Fatigue time of 1-week-old flies in TOR^UAS-RNAi group. (N) Fatigue time of 3-week-old flies in TOR^UAS-RNAi group. (O) Fatigue time of flies aged 4 weeks in TOR^UAS-RNAi group. (P) Fatigue time of 5-week-old flies in TOR^UAS-RNAi group. (Q) Body weight of 5-week-old flies in TOR^UAS-RNAi group. (R) Climb height of 1,3,4,5 week-old flies in TOR^UAS-RNAi group within 3 seconds. (S) Skeletal muscle ROS levels in TOR^UAS-RNAi group. (T) SOD level in skeletal muscle of TOR^UAS-RNAi group. (U) Relative expression of TOR gene in skeletal muscle of TOR^UAS-RNAi group. (V) Relative expression of ATG2 gene in skeletal muscle of TOR^UAS-RNAi group. (W) Relative expression of PGC-1α gene in skeletal muscle of TOR^UAS-RNAi group. (X) Relative expression of SIR2 gene in skeletal muscle of TOR^UAS-RNAi group. (Y) myosin heavy chain immunofluorescence in TOR^UAS-OE group, with blue dots as nucleus and red fluorescent bands as myofibril. HSD reduces fluorescence expression (scale: white line is 5 microns). (Z) TOR^UAS-RNAi group myosin heavy chain immunofluorescence. (α) Skeletal muscle transmission electron microscopy (scale: small mesh represents 0.2 microns). The white arrow points to the myofibril. The red arrow points to the mitochondria. Climbing height measurements, the sample size of each group is about 150–170 animals. Univariate analysis of variance (ANOVA) and minimum significance difference (LSD) tests were used to determine differences between groups. The P-values of climbing endurance curve and survival curve were calculated by log-rank test. Data were expressed by mean ±SEM. * $P < 0.05$; * * $P < 0.01$; * * * $P < 0.001$; n indicates no significant difference. The sample size of proteins and ROS was muscle of 20 flies per group, measured 3 times. The sample size for RT-PCR and ELISA was muscle of 20 flies per group, measured 3 times. Univariate analysis of variance (ANOVA) and minimum significance difference (LSD) tests were used to determine differences between groups. Data were expressed by mean ±SEM. * $P < 0.05$; * * $P < 0.01$; n indicates no significant difference.

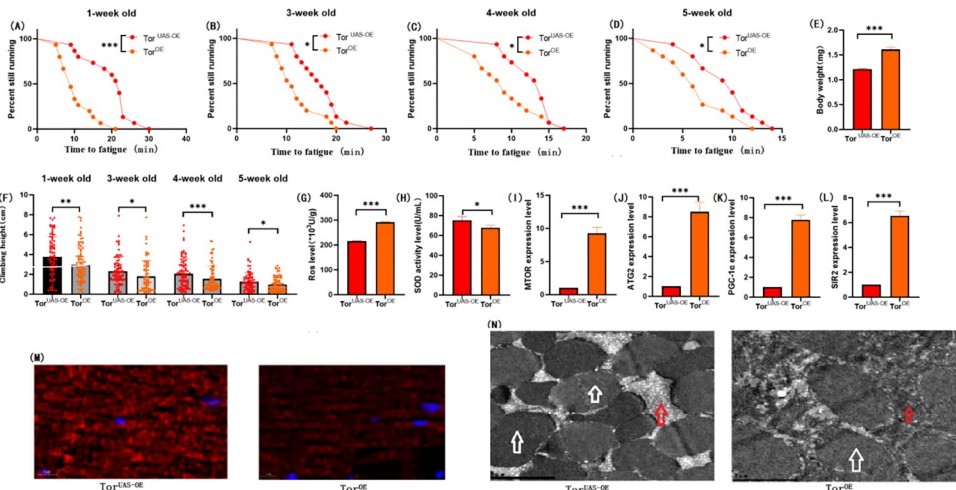

**Fig 6. Effects of TOR overexpression on body weight, climbing ability and skeletal muscle physiological structure of TOR^UAS-OE flies'.** (A) Fatigue time of 1-week-old flies. (B) Fatigue time of flies at 3 weeks of age. (C) Fatigue time of flies at 4 weeks of age. (D) Fatigue time of 5 week-old flies. (E) Body weight of flies at 5 weeks of age. (F) Climb height of 1,3,4,5 week-old flies within 3 seconds. (G) Skeletal muscle ROS levels. (H) SOD level in skeletal muscle. (I) Relative expression of TOR gene in skeletal muscle. (J) Relative expression of skeletal muscle ATG2 gene. (K) Relative expression of PGC-1α gene in skeletal muscle. (L) Relative expression of SIR2 gene in skeletal muscle. (M) myosin heavy chain immunofluorescence in TOR^UAS-OE group, with blue dots as nucleus and red fluorescent bands as myofibril. HSD reduces fluorescence expression (scale: white line is 5 microns). (N) Skeletal muscle transmission electron microscopy (scale: small mesh represents 0.2 microns). The white arrow points to the myofibril. The red arrow points to the mitochondria. Climbing height measurements, the sample size of each group is about 150–170 animals. Univariate analysis of variance (ANOVA) and minimum significance difference (LSD) tests were used to determine differences between groups. The P-values of climbing endurance curve and survival curve were calculated by log-rank test. Data were expressed by mean ±SEM. * P < 0.05; * * P < 0.01; * * * P < 0.001; n indicates no significant difference. The sample size of proteins and ROS was muscle of 20 flies per group, measured 3 times. The sample size for RT-PCR and ELISA was muscle of 20 flies per group, measured 3 times. (A) Univariate analysis of variance (ANOVA) and minimum significance Difference (LSD) tests were used to determine inter-group differences. Data were expressed by mean ±SEM. * P < 0.05; * * P < 0.01; n indicates no significant difference.

In muscular TOR^OE Drosophila, HSD significantly reduced the expressions of the MTOR gene (Fig 8I; P < 0.01), ATG2 gene (Fig 8J; P < 0.01), PGC-1α gene (Fig 8K; P < 0.001), and SIR2 gene (Fig 8L; P < 0.01), but had no significant effect on SOD activity in muscle (Fig 8H; P > 0.05). Muscle ROS levels were significantly increased (Fig 8G; P < 0.05). Myosin heavy chain red immunofluorescence images showed that TOR^OE and HSD reduced myosin heavy chain fluorescence expression and muscle fiber alignment(Fig 8M), and Tor knockdown reversed this change (Fig 12M). Skeletal muscle transmission electron microscopy images suggest that TOR knockdown can maintain the normal morphology and structure of myofibrillar fibers and mitochondria in aging skeletal muscle of TOR^OE (Fig 6N).

## 3.4 Muscular TOR knockdown protects skeletal muscle from HSD-induced age-related degeneration

By constructing the mhc-Gal4/ Tor-UAS-RNAi (TOR^RNAi) system to knockdown the TOR gene in Drosophila skeletal muscle, the role of the muscle TOR/ATG2/PGC-1α pathway and the TOR/SIR2/PGC-1α pathway in exercise against HSD-induced skeletal muscle aging was further investigated. The results showed that RNAi of the muscle TOR gene had no significant effect on Drosophila body weight (Fig 10E; P > 0.05), and exercise and HSD reduced the body weight of TOR^RNAi Drosophila (Figs 11E, 12E; P < 0.05). Muscle TOR^RNAi had no significant effect on the climbing height of flies at 1 week of age (Fig 10F; P > 0.05), but muscle TOR^RNAi

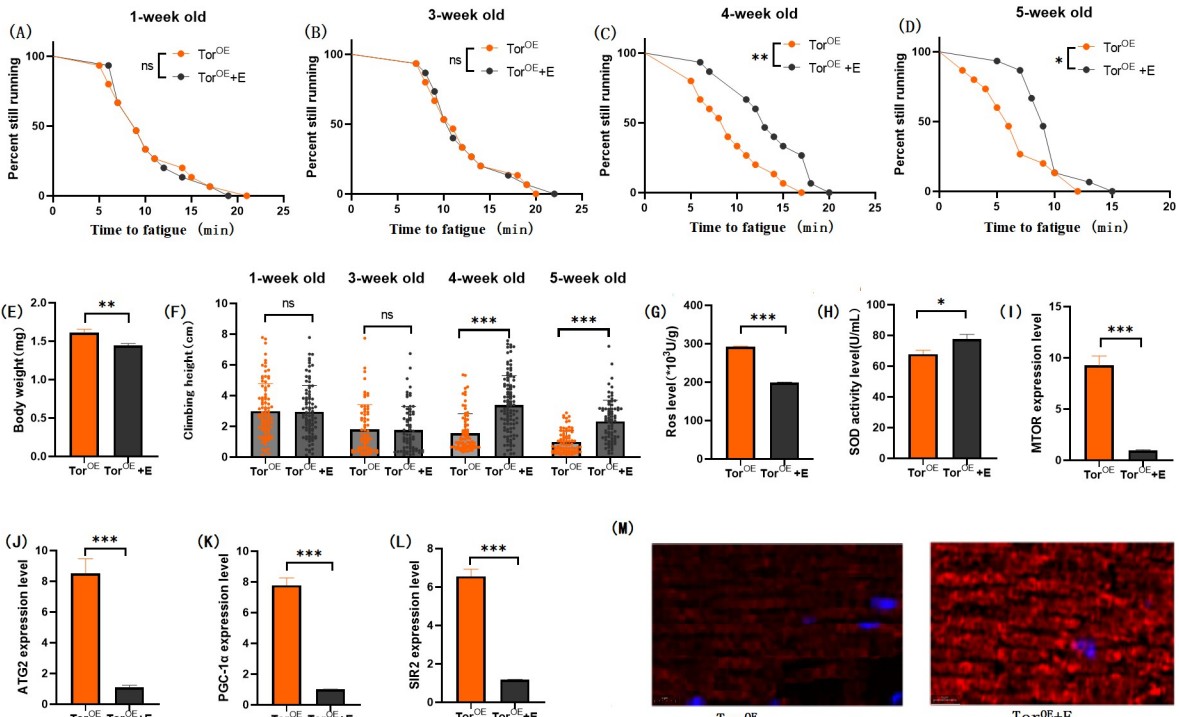

**Fig 7. Effects of exercise on body weight, climbing ability and skeletal muscle physiological structure of TOR^OE flies.** (A) Fatigue time of 1-week-old flies. (B) Fatigue time of flies at 3 weeks of age. (C) Fatigue time of flies at 4 weeks of age. (D) Fatigue time of 5 week-old flies. (E) Body weight of flies at 5 weeks of age. (F) Climb height of 1,3,4,5 week-old flies within 3 seconds. (G) Skeletal muscle ROS levels. (H) SOD level in skeletal muscle. (I) Relative expression of TOR gene in skeletal muscle. (J) Relative expression of skeletal muscle ATG2 gene. (K) Relative expression of PGC-1α gene in skeletal muscle. (L) Relative expression of SIR2 gene in skeletal muscle. (M) myosin heavy chain immunofluorescence in TOR^UAS-OE group, with blue dots as nucleus and red fluorescent bands as myofibril. HSD reduces fluorescence expression (scale: white line is 5 microns). Climbing height measurements, the sample size of each group is about 150–170 animals. Univariate analysis of variance (ANOVA) and minimum significance difference (LSD) tests were used to determine differences between groups. The P-values of climbing endurance curve and survival curve were calculated by log-rank test. Data were expressed by mean ±SEM. * P < 0.05; * * P < 0.01; * * * P < 0.001; n indicates no significant difference. The sample size of proteins and ROS was muscle of 20 flies per group, measured 3 times. The sample size for RT-PCR and ELISA was muscle of 20 flies per group, measured 3 times. (A) Univariate analysis of variance (ANOVA) and minimum significance Difference (LSD) tests were used to determine inter-group differences. Data were expressed by mean ±SEM. * P < 0.05; * * P < 0.01; n indicates no significant difference.

significantly increased the climbing height of flies at 3, 4, and 5 weeks of age (Fig 10F; P < 0.001). Exercise significantly increased the climbing height of TOR^RNAi flies and TOR^RNAi + HSD flies (Figs 11F, 13F; P < 0.05 or P < 0.001). HSD significantly reduced the climbing height of TOR^RNAi flies (Fig 12F; P < 0.01 or P < 0.001).

The results showed that TOR^RNAi had no significant effect on CTF of 1-week-old flies (Fig 10A; P > 0.05), but significantly increased CTF in flies of 3, 4, and 5 weeks of age (Fig 10B–10D; P < 0.01 or P < 0.001). Exercise significantly increased CTF in TOR^RNAi and TOR^RNAi + HSD flies (Figs 11A–11D, 12A–12D; P < 0.05 or P < 0.01 or P < 0.001). HSD significantly reduced CTF in TOR^RNAi flies (Fig 12A–12D; P < 0.01 or P < 0.001).

In addition, TOR^RNAi significantly reduced the expression of muscle MTOR genes (Fig 10I; P<0.001), ATG2 genes (Fig 10J; P < 0.001), PGC-1α genes (Fig 10K; P < 0.001), and SIR2 genes (Fig 10L; P < 0.001), and significantly reduced muscle ROS levels (Fig 10G; P < 0.001). However, SOD activity in muscle was significantly increased (Fig 10H; P < 0.05). Myosin heavy chain red immunofluorescence images showed that TOR^RNAi increased myosin heavy chain fluorescence expression and muscle fiber alignment (Fig 10M).

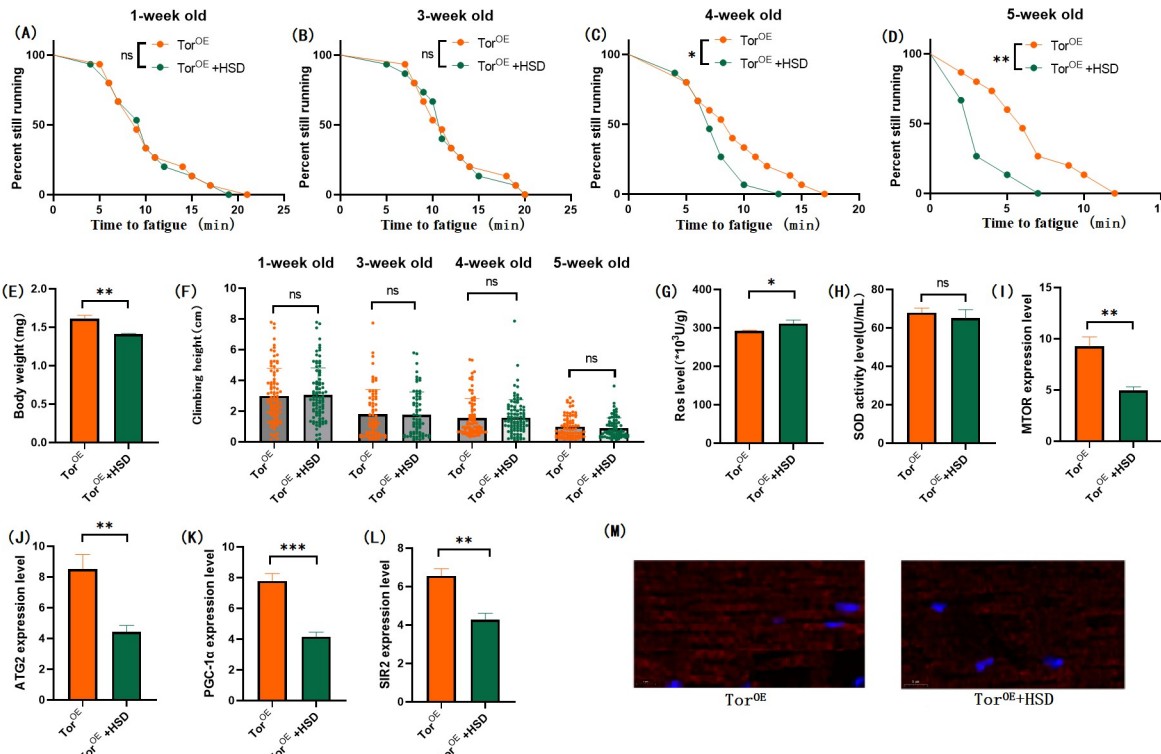

**Fig 8. Effects of HSD on body weight, climbing ability and skeletal muscle physiological structure of TOR^OE flies.** (A) Fatigue time of 1-week-old flies. (B) Fatigue time of flies at 3 weeks of age. (C) Fatigue time of flies at 4 weeks of age. (D) Fatigue time of 5 week-old flies. (E) Body weight of flies at 5 weeks of age. (F) Climb height of 1,3,5 week-old flies within 3 seconds. (G) Skeletal muscle ROS levels. (H) SOD level in skeletal muscle. (I) Relative expression of TOR gene in skeletal muscle. (J) Relative expression of skeletal muscle ATG2 gene. (K) Relative expression of PGC-1α gene in skeletal muscle. (L) Relative expression of SIR2 gene in skeletal muscle. (M) myosin heavy chain immunofluorescence in TOR^UAS-OE group, with blue dots as nucleus and red fluorescent bands as myofibril. HSD reduces fluorescence expression (scale: white line is 5 microns). Climbing height measurements, the sample size of each group is about 150–170 animals. Univariate analysis of variance (ANOVA) and minimum significance difference (LSD) tests were used to determine differences between groups. The P-values of climbing endurance curve and survival curve were calculated by log-rank test. Data were expressed by mean ±SEM.* P < 0.05; * * P < 0.01; * * * P < 0.001; n indicates no significant difference. The sample size of proteins and ROS was muscle of 20 flies per group, measured 3 times. The sample size for RT-PCR and ELISA was muscle of 20 flies per group, measured 3 times. (A) Univariate analysis of variance (ANOVA) and minimum significance Difference (LSD) tests were used to determine inter-group differences. Data were expressed by mean ±SEM. * P < 0.05; * * P < 0.01; n indicates no significant difference.

In muscle TOR^RNAi fly, HSD significantly increased the expression of the MTOR gene (Fig 12I; P < 0.001), ATG2 gene (Fig 12J; P < 0.001), PGC-1α gene (Fig 12K; P < 0.001), and SIR2 gene (Fig 12L; P < 0.01), but had no significant effect on muscle SOD activity (Fig 12H; P > 0.05). Muscle ROS levels were significantly increased (Fig 12G; P < 0.01). Myosin heavy chain red immunofluorescence images showed that HSD reduced myosin heavy chain fluorescence expression and muscle fiber alignment (Fig 12M).

In TOR^RNAi Drosophila and TOR^RNAi + HSD Drosophila muscles, exercise did not significantly change the muscle SIR2 gene (Fig 13L; P > 0.05), but significantly reduced the muscle MTOR gene (Fig 13I; P < 0.01), ATG2 gene (Fig 13J; P < 0.01), PGC-1α gene (Fig 13K; P < 0.01). SOD activity in muscle was significantly increased (Fig 13H; P < 0.001), and ROS level in muscle was significantly decreased (Fig 13G; P < 0.01). Myosin heavy chain red immunofluorescence images showed that HSD reduced myosin heavy chain fluorescence expression and muscle fiber alignment, and exercise combined with TOR gene knockdown could resist or even reverse it (Fig 12M).

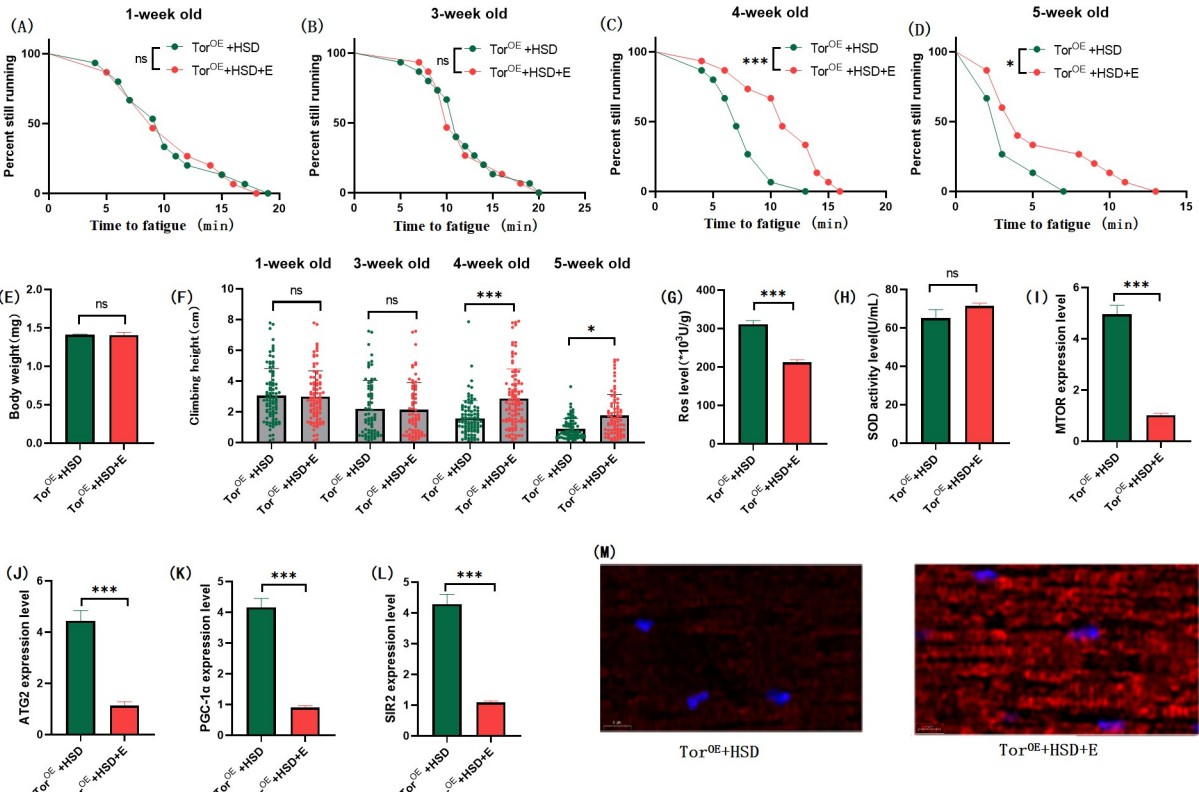

**Fig 9. Effects of exercise and HSD on body weight, climbing ability and skeletal muscle physiological structure of TOR$^{OE}$ flies.** (A) Fatigue time of 1-week-old flies. (B) Fatigue time of flies at 3 weeks of age. (C) Fatigue time of flies at 4 weeks of age. (D) Fatigue time of 5 week-old flies. (E) Body weight of flies at 5 weeks of age. (F) Climb height of 1,3,4,5 week-old flies within 3 seconds. (G) Skeletal muscle ROS levels. (H) SOD level in skeletal muscle. (I) Relative expression of TOR gene in skeletal muscle. (J) Relative expression of skeletal muscle ATG2 gene. (K) Relative expression of PGC-1α gene in skeletal muscle. (L) Relative expression of SIR2 gene in skeletal muscle. (M) myosin heavy chain immunofluorescence in TOR$^{UAS-OE}$ group, with blue dots as nucleus and red fluorescent bands as myofibril. HSD reduces fluorescence expression (scale: white line is 5 microns). Climbing height measurements, the sample size of each group is about 150–170 animals. Univariate analysis of variance (ANOVA) and minimum significance difference (LSD) tests were used to determine differences between groups. The P-values of climbing endurance curve and survival curve were calculated by log-rank test. Data were expressed by mean ±SEM. * P < 0.05; * * P < 0.01; * * * P < 0.001; n indicates no significant difference. The sample size of proteins and ROS was muscle of 20 flies per group, measured 3 times. The sample size for RT-PCR and ELISA was muscle of 20 flies per group, measured 3 times. (A) Univariate analysis of variance (ANOVA) and minimum significance Difference (LSD) tests were used to determine inter-group differences. Data were expressed by mean ±SEM. * P < 0.05; * * P < 0.01; n indicates no significant difference.

## 4 Discussion

The risk of sarcopenia, which is associated with decreased muscle mass and strength, is associated with salt intake [1, 3]. Studies in fruit flies have shown that HSD negatively affects age-related climbing ability and mortality by up-regulating CG2196(SALT) expression and inhibiting the dFOXO/SOD pathway, and that increased dFOXO/SOD pathway activity plays a key role in mediating endurance exercise resistance to low salt tolerance [4, 5], resulting in impaired climbing ability and longevity in aging fruit flies. A high-salt diet induces muscle autophagy and activates TORC1. Studies on rats have shown that activation of TORC1 can lead to salt-induced hypertension and kidney damage in dahl salt-sensitive (SS) rats [19]. As a transcriptional coactivator of many genes, PGC-1α is involved in energy metabolism management and mitochondrial biogenesis, and the expression of PGC-1α is closely related to body senescence, cell senescence, and many age-related diseases [20]. In this experiment, because the trend of the PCR gene expression map was basically the same, it was impossible to

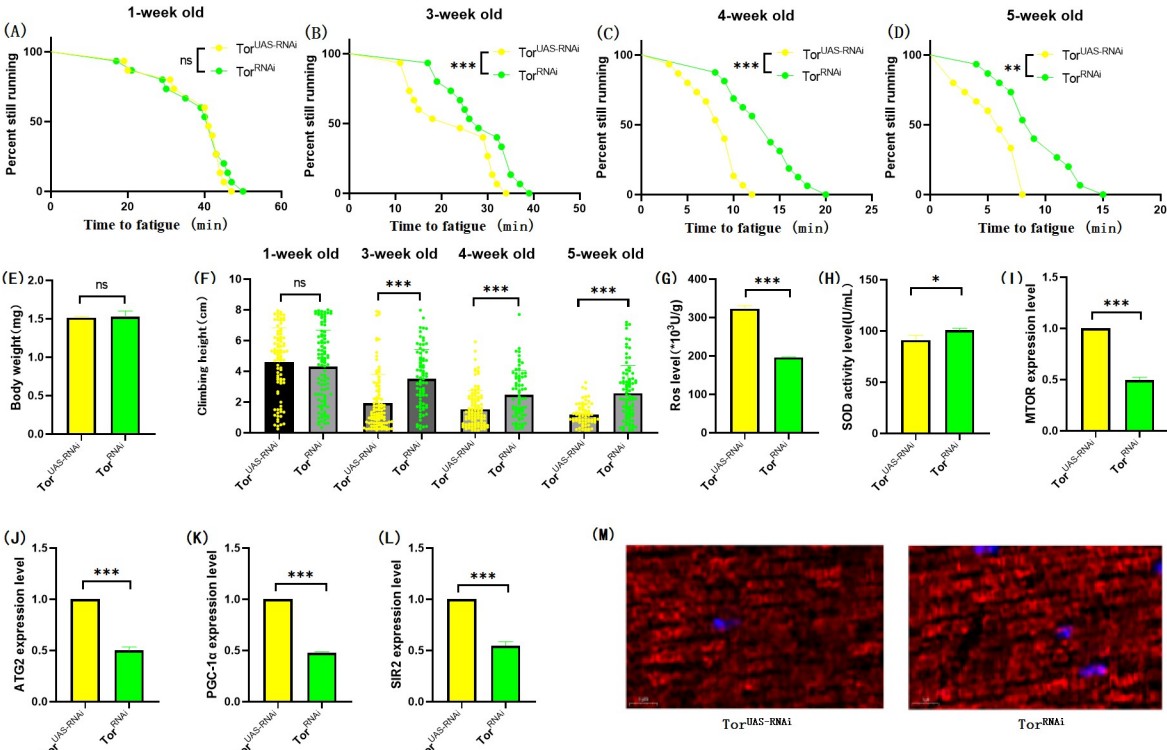

**Fig 10. Effects of TOR knockdown on body weight, climbing ability and skeletal muscle physiological structure of TOR^UAS-RNAi flies.**
(A) Fatigue time of 1-week-old flies. (B) Fatigue time of flies at 3 weeks of age. (C) Fatigue time of flies at 4 weeks of age. (D) Fatigue time of 5 week-old flies.(E) Body weight of flies at 5 weeks of age. (F) Climb height of 1,3,4,5 week-old flies within 3 seconds. (G) Skeletal muscle ROS levels. (H) SOD level in skeletal muscle. (I) Relative expression of TOR gene in skeletal muscle. (J) Relative expression of skeletal muscle ATG2 gene. (K) Relative expression of PGC-1α gene in skeletal muscle. (L) Relative expression of SIR2 gene in skeletal muscle. (M) myosin heavy chain immunofluorescence in TOR^UAS-OE group, with blue dots as nucleus and red fluorescent bands as myofibril. HSD reduces fluorescence expression (scale: white line is 5 microns). Climbing height measurements, the sample size of each group is about 150–170 animals. Univariate analysis of variance (ANOVA) and minimum significance difference (LSD) tests were used to determine differences between groups. The P-values of climbing endurance curve and survival curve were calculated by log-rank test. Data were expressed by mean ±SEM. * P < 0.05; * * P < 0.01; * * * P < 0.001; n indicates no significant difference. The sample size of proteins and ROS was muscle of 20 flies per group, measured 3 times. The sample size for RT-PCR and ELISA was muscle of 20 flies per group, measured 3 times. (A) Univariate analysis of variance (ANOVA) and minimum significance Difference (LSD) tests were used to determine inter-group differences. Data were expressed by mean ±SEM. * P < 0.05; * * P < 0.01; n indicates no significant difference.

intuitively see the rise and fall of each gene expression level. Therefore, the average value of the gene expression level of each group in a pair-to-pair comparison was compared (the relative expression ratio of a single gene = the gene expression level of the intervention group/the gene expression level of the control group).However, through pairwise comparison of MTOR gene expression levels in overexpressed/knockdown flies aged three weeks, we determined that the overexpressed/knockdown system was successfully constructed.The relative gene expression ratio of the HSD group and the gene control group was found to inhibit ATG2, PGC-1α, and SIR2. We speculate that HSD may cause MTOR/SIR2/PGC-1α pathway inhibition. SOD is an oxygen free radical scavenging enzyme [21]. It was found that the SIR2 homologous gene SSIRT1126 could be activated by resveratrol to increase SOD activity, decrease ROS levels, and increase PGC-1 expression [22]. One of the main functions of PGC-1α is as a ROS-scavenging enzyme regulator [23].Therefore, the inhibitory effect of HSD on the MTOR/SIR2/PGC-1α pathway may be one of the ways to increase the oxidative damage of skeletal muscle cells, and the up-regulation of autophagy by rapamycin (a TOR inhibiTOR) can reduce ATG2 and PGC-1α induced aging [24]. Therefore, the inhibitory effect of HSD on the MTOR/ATG2/

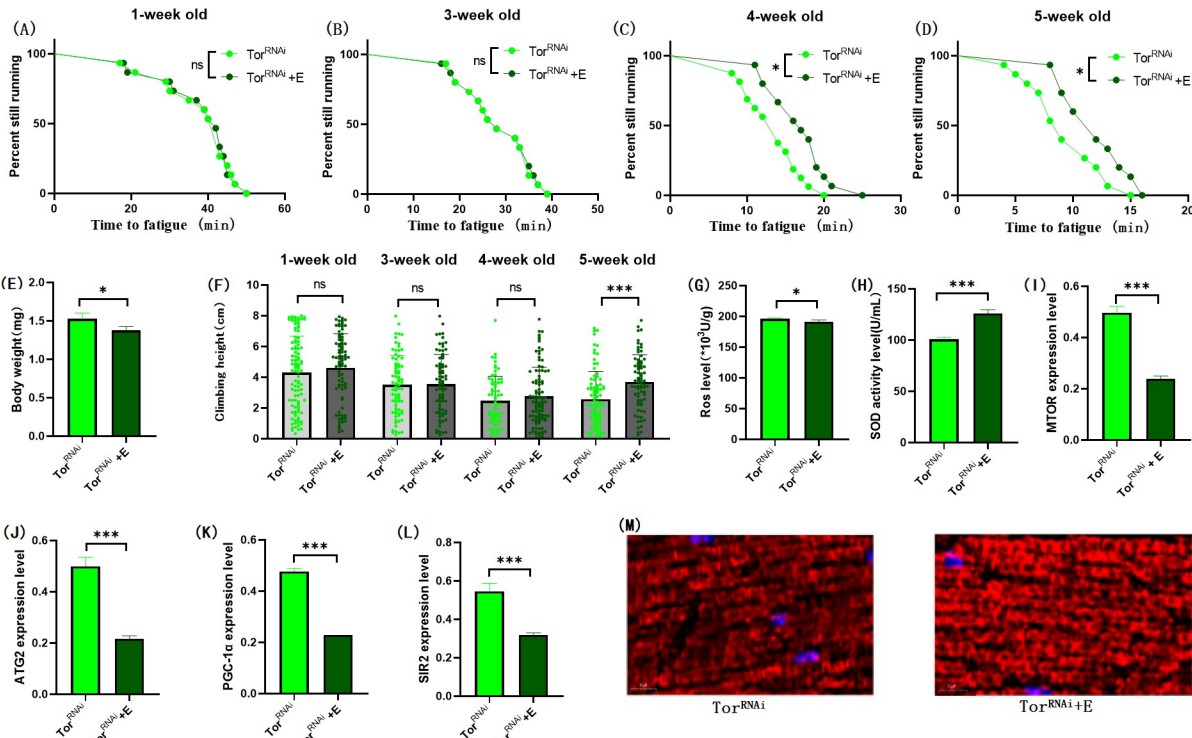

**Fig 11. Effects of exercise on body weight, climbing ability and skeletal muscle physiological structure of TOR$^{RNAi}$ flies.** (A) Fatigue time of 1-week-old flies. (B) Fatigue time of flies at 3 weeks of age. (C) Fatigue time of flies at 4 weeks of age. (D) Fatigue time of 5 week-old flies. (E) Body weight of flies at 5 weeks of age. (F) Climb height of 1,3,4,5 week-old flies within 3 seconds. (G) Skeletal muscle ROS levels. (H) SOD level in skeletal muscle. (I) Relative expression of TOR gene in skeletal muscle. (J) Relative expression of skeletal muscle ATG2 gene. (K) Relative expression of PGC-1α gene in skeletal muscle. (L) Relative expression of SIR2 gene in skeletal muscle. (M) myosin heavy chain immunofluorescence in TOR$^{UAS-OE}$ group, with blue dots as nucleus and red fluorescent bands as myofibril. HSD reduces fluorescence expression (scale: white line is 5 microns). Climbing height measurements, the sample size of each group is about 150–170 animals. Univariate analysis of variance (ANOVA) and minimum significance difference (LSD) tests were used to determine differences between groups. The P-values of climbing endurance curve and survival curve were calculated by log-rank test. Data were expressed by mean ±SEM.* P < 0.05; * * P < 0.01; * * * P < 0.001; n indicates no significant difference. The sample size of proteins and ROS was muscle of 20 flies per group, measured 3 times. The sample size for RT-PCR and ELISA was muscle of 20 flies per group, measured 3 times. (A) Univariate analysis of variance (ANOVA) and minimum significance Difference (LSD) tests were used to determine inter-group differences. Data were expressed by mean ±SEM. * P < 0.05; * * P < 0.01; n indicates no significant difference.

PGC-1α pathway may be one of the ways to increase the oxidative damage of skeletal muscle cells.

Although these results suggest that HSD may play a role in inhibiting the MTOR/SIR2/ PGC-1α and MTOR/ATG2/PGC-1α pathways, but it is unclear whether the MTOR/SIR2/ PGC-1α and MTOR/ATG2/PGC-1α pathways can modulate the effects of HSD on skeletal muscle aging. To confirm this, the UAS/mhcGal4 system was used to construct the differential expression of TOR gene in Drosophila muscle. In Drosophila melanogaster, the expression of multiple muscle genes can delay the muscle aging caused by HSD, such as skeletal muscle PGC-1α, ATG2 and SIR2 [25]. On the contrary, knockdown of these genes will accelerate the aging of skeletal muscle and increase the damage of skeletal muscle caused by oxidative stress [18, 25]. Skeletal muscle PGC-1α, ATG2 and SIR2 have been identified as key antagonists of HSD-induced Drosophila senescence [24]. These studies also show that differential expression of genes has become a practical way to confirm gene function. Studies on TOR signaling in muscles suggest that TORC1 activation may cause harmful accumulation due to aging [26]. Recent studies have shown that the downregulation of TOR signaling, together with the

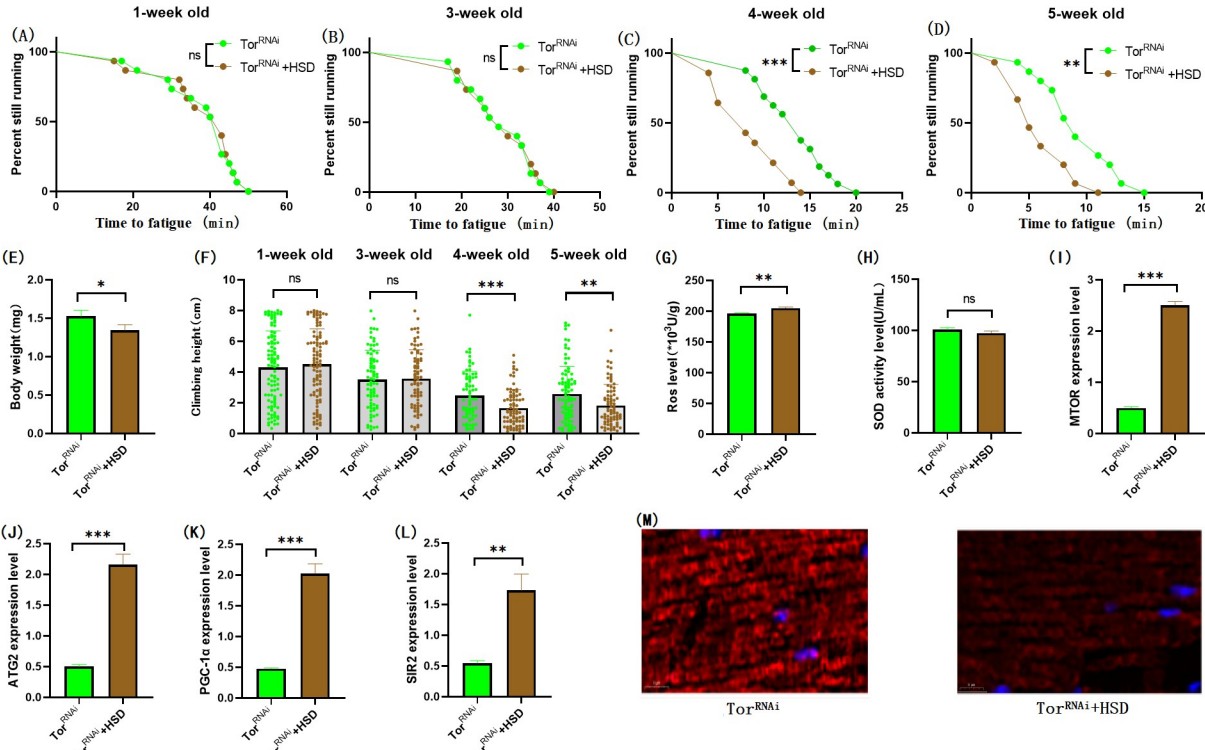

**Fig 12. Effects of HSD on body weight, climbing ability and skeletal muscle physiological structure of TOR$^{RNAi}$ flies.** (A) Fatigue time of 1-week-old flies. (B) Fatigue time of flies at 3 weeks of age. (C) Fatigue time of flies at 4 weeks of age. (D) Fatigue time of 5 week-old flies. (E) Body weight of flies at 5 weeks of age. (F) Climb height of 1,3,4,5 week-old flies within 3 seconds. (G) Skeletal muscle ROS levels. (H) SOD level in skeletal muscle. (I) Relative expression of TOR gene in skeletal muscle. (J) Relative expression of skeletal muscle ATG2 gene. (K) Relative expression of PGC-1α gene in skeletal muscle. (L) Relative expression of SIR2 gene in skeletal muscle. (M) myosin heavy chain immunofluorescence in TOR$^{UAS-OE}$ group, with blue dots as nucleus and red fluorescent bands as myofibril.HSD reduces fluorescence expression (scale: white line is 5 microns). Climbing height measurements, the sample size of each group is about 150–170 animals. Univariate analysis of variance (ANOVA) and minimum significance difference (LSD) tests were used to determine differences between groups. The P-values of climbing endurance curve and survival curve were calculated by log-rank test. Data were expressed by mean ±SEM.* P < 0.05; * * P < 0.01; * * * P < 0.001; n indicates no significant difference. The sample size of proteins and ROS was muscle of 20 flies per group, measured 3 times. The sample size for RT-PCR and ELISA was muscle of 20 flies per group, measured 3 times. (A) Univariate analysis of variance (ANOVA) and minimum significance Difference (LSD) tests were used to determine inter-group differences. Data were expressed by mean ±SEM. * P < 0.05; * * P < 0.01; n indicates no significant difference.

stimulation of autophagy, can update muscle structure and improve the overall quality of organelles, thereby inhibiting the aging of skeletal muscle [26]. In this study, compared with the TOR$^{UAS-OE}$ group, the relative expression ratio of other genes except TOR decreased in the TOR$^{OE}$ group, indicating that the expression level of other genes in the TOR$^{OE}$ group decreased. Therefore, we seem to think that TOR activation in muscle inhibits the MTOR/SIR2/PGC-1α and MTOR/ATG2/PGC-1α pathways in muscle, reducing the antioxidant stress capacity of skeletal muscle. In this experiment, we found that TOR gene knockout had a completely opposite effect, and TOR knockdown activated the MTOR/SIR2/PGC-1α and MTOR/ATG2/PGC-1α pathways, improving the antioxidant stress ability of skeletal muscle. These results suggest that activation of the MTOR/SIR2/PGC-1α and TOR/ATG2/PGC-1α pathways may play an important role in inhibiting skeletal muscle aging. To further confirm this idea, TOR knockdown flies were fed HSD. The results showed that the relative expression ratio of the TOR gene in the TOR$^{RNAi}$ + HSD group was higher than that of other genes in the TOR$^{RNAi}$ + HSD group, indicating that the expression level of other genes in the TOR$^{RNAi}$ + HSD group was reduced. Therefore, HSD decreased the activity of the MTOR/SIR2/PGC-1α

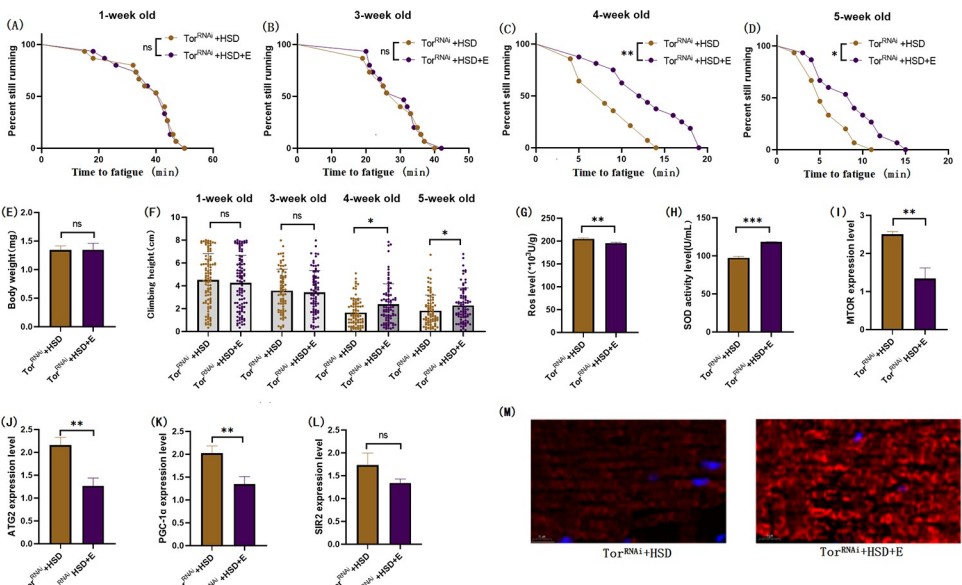

**Fig 13. Effects of exercise and HSD on body weight, climbing ability and skeletal muscle physiological structure of TOR^RNAi flies.** (A) Fatigue time of 1-week-old flies. (B) Fatigue time of flies at 3 weeks of age. (C) Fatigue time of flies at 4 weeks of age. (D) Fatigue time of 5 week-old flies. (E) Body weight of flies at 5 weeks of age. (F) Climb height of 1,3,4,5 week-old flies within 3 seconds. (G) Skeletal muscle ROS levels. (H) SOD level in skeletal muscle. (I) Relative expression of TOR gene in skeletal muscle. (J) Relative expression of skeletal muscle ATG2 gene. (K) Relative expression of PGC-1α gene in skeletal muscle. (L) Relative expression of SIR2 gene in skeletal muscle. (M) myosin heavy chain immunofluorescence in TOR^UAS-OE group, with blue dots as nucleus and red fluorescent bands as myofibril. HSD reduces fluorescence expression (scale: white line is 5 microns). Climbing height measurements, the sample size of each group is about 150–170 animals. Univariate analysis of variance (ANOVA) and minimum significance difference (LSD) tests were used to determine differences between groups. The P-values of climbing endurance curve and survival curve were calculated by log-rank test. Data were expressed by mean ±SEM.* P < 0.05; * * P < 0.01; * * * P < 0.001; n indicates no significant difference. The sample size of proteins and ROS was muscle of 20 flies per group, measured 3 times. The sample size for RT-PCR and ELISA was muscle of 20 flies per group, measured 3 times. (A) Univariate analysis of variance (ANOVA) and minimum significance Difference (LSD) tests were used to determine inter-group differences. Data were expressed by mean ±SEM. * P < 0.05; * * P < 0.01; n indicates no significant difference.

and MTOR/ATG2/PGC-1α pathways in TOR knockout Drosophila melanogaster, and decreased the ability of skeletal muscle to resist oxidative stress. In addition, HSD had a negative effect on the climbing endurance and climbing height of TOR knockout Drosophila. These results demonstrate that TOR gene knockdown does not seem to be able to counteract the negative effects of HSD on skeletal muscle aging.

Exercise can alleviate the negative effects of sarcopenia caused by HSD to a certain extent [3]. Long-term exercise has also been shown to induce pleiotropic effects in favor of aging muscle, including immunosuppression, enhanced muscle oxidation capacity, and a lean phenotype [27]. Another study showed that endurance exercise improved climbing ability and survival in HSD flies [5]. Studies on the elderly found that aerobic exercise increased the antioxidant SOD level, and, at the same time, improved the oxidative stress damage caused by ROS [3]. Although endurance exercise activates PGC-1α and enhances skeletal muscle mitochondrial function, it is unclear whether exercise can up-regulate the MTOR/SIR2/PGC-1α and MTOR/ATG2/PGC-1α pathways. In this study, we found that endurance exercise can improve oxidative stress damage to skeletal muscle from HSD, which is consistent with previous studies. In addition, we also found that exercise can activate the SIR2 gene, and exercise can reverse the age-related changes caused by Atg2 knockout, thus playing a certain role in

delaying aging. In this study, compared with the HSD group, the relative expression ratios of ATG2, SIR2, and PGC-1α genes in the HSD combined exercise group were all increased, indicating that exercise under the intervention of HSD can improve the activity levels of ATG2, PGC-1α, and SIR2 genes. Therefore, exercise can improve the inhibitory effect of HSD on TOR/SIR2/PGC-1α and TOR/ATG2/PGC-1α pathways, which may be the molecular mechanism of exercise against HSD-induced skeletal muscle aging.

To confirm the hypothesis, we performed a motor intervention on TOR-overexpressing Drosophila melanogaster. We found that endurance exercise improved skeletal muscle aging caused by TOR gene overexpression but did not reduce the expression of the TOR gene in muscle. Endurance exercise can activate myocardial AMP-activated protein kinase (AMPK), and the activated AMPK can effectively inhibit TOR activity [28]. Recent studies have found that inhibition of TOR can regulate senescence caused by ATG2 and PGC-1α knockdown through up-regulation of autophagy [25]. Therefore, we speculate that exercise activates ATG2 and PGC-1α by inhibiting skeletal muscle TOR activity, so as to achieve the effect of resisting skeletal muscle aging. Exercise can improve the antioxidant stress capacity of skeletal muscle by activating PGC-1α and SIR2, and exercise can also improve aging caused by ATG2 knockdown [25]. These evidences may account for the activation of the TOR/SIR2/PGC-1α and TOR/ATG2/PGC-1α pathways by exercise. This hypothesis needs further study. Current evidence suggests that the MTOR/SIR2/PGC-1α pathway and the MTOR/ATG2/PGC-1α pathway are important molecular mechanisms of resistance to skeletal muscle aging during endurance exercise. In addition, exercise has the role of regulating and controlling weight [29], which is similar to the results of this study. In this study, both pathways activated by exercise seem to play a good role in regulating body weight.

We also investigated the relationship between TOR, HSD, and exercise on activity and skeletal muscle aging in Drosophila melanogaster. The aging of skeletal muscle is accompanied by a decrease in mitochondrial content and function, an increase in ROS levels, and a decrease in anti-oxidative stress ability of skeletal muscle [30]. TOR knockdown can delay the aging of skeletal muscle by enhancing the antioxidant stress ability, improving motor activity and regulating the autophagy of fruit flies [31]. In this study, we found that overexpression of TOR led to a decrease in Drosophila motility and reduced the antioxidant stress ability of Drosophila. Relevant studies have proved that TOR gene knockout has a positive effect on weight loss [32], but there are few studies on the effect of TOR activation on body weight. In this study, TOR activation leads to weight gain in Drosophila melanogaster, which may provide a theoretical basis for future related studies. In addition, the expression of TOR also promotes the aging of skeletal muscle, which is achieved by inhibiting ATG2 to delay autophagy [33]. This was basically consistent with the results of our current study. Exercise activated the MTOR/ATG2/PGC-1α and MTOR/SIR2/PGC-1α pathways, increased SOD activity in Drosophila skeletal muscle, decreased ROS activity, and inhibited oxidative stress damage to skeletal muscle cells. We also found that exercise significantly increased fatigue time and climbing height in flies. In this study, we can find out that exercise improved the decline of skeletal muscle anti-oxidative stress ability caused by inhibition of MTOR/ATG2/PGC-1α and MTOR/SIR2/PGC-1α pathways induced by high salt, and increased the fatigue time and climbing height of flies with high salt. Finally, the negative effects of TOR gene overexpression and HSD on skeletal muscle can also be resisted by exercise. Exercise combined with TOR gene knockdown can better combat the damage of HSD to skeletal muscle.

## 5 Conclusion

Muscle TOR knocking down and endurance exercise can improve skeletal muscle aging to some extent, and exercise can also resist the skeletal muscle aging caused by HSD, while

muscle TOR knocking down does not seem to have a significant improvement on the skeletal muscle aging caused by HSD. Better anti-aging effects can be achieved when the two are combined. TOR overexpression and HSD can accelerate muscle tissue damage and accelerate skeletal muscle aging. The combination of the two causes more damage to skeletal muscles, but exercise can improve the situation to some extent. The molecular mechanism of these processes is related to the effects of muscle TOR gene and exercise on the MTOR/ATG2/PGC-1α, MTOR/SIR2/PGC-1α pathway and oxidation balance.

## Supporting information

**S1 File.**
(XLSX)

## Acknowledgments

Thanks to Professor wen dt for the ideas provided for this research, and thanks to Laboratory 303, School of Physical Education, Ludong University for the technical support of this research.

## Author Contributions

**Data curation:** Jing-feng Wang, Xing-feng Ma.

**Supervision:** Ying-hui Gao.

**Writing – original draft:** Shi-jie Wang.

**Writing – review & editing:** Deng-tai Wen.

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
