## [Decision Letter · Decision Letter 0]

23 Jul 2024

PONE-D-24-20830Muscular TOR knockdown and endurance exercise delayed age-related deterioration of skeletal muscle by activating the SIR2/PGC-1α and ATG2/PGC-1α pathway in Drosophila on a high-salt dietPLOS ONE

Dear Dr. Wang,

Thank you for submitting your manuscript to PLOS ONE. After careful consideration, we feel that it has merit but does not fully meet PLOS ONE’s publication criteria as it currently stands. Therefore, we invite you to submit a revised version of the manuscript that addresses the points raised during the review process.

The reviews for the above manuscript have been received. The reviewers' comments are included along with this letter. Three independent reviewers have assessed the manuscript's suitability for publication and solicit additional clarification on certain outstanding issues. 

While your study offers valuable insights into sarcopenia mechanisms, several critical issues must be addressed. Firstly, the rationale for using flies with TOR knock-down or overexpression requires further elucidation, along with inclusion of data from wild type flies and validation of TOR RNAi efficiency. Additionally, the non-specific expression of MHC-GAL4 outside skeletal muscles warrants clarification with a more specific driver, supported by foundational experiment replication.

If you wish to revise the manuscript to address the issues raised by reviewers, editors would be happy to consider the revised version of the manuscript for publication. 

When revising the manuscript, please consider all the points raised by the reviewers and outline every change made. If you disagree with the reviewers' comments, please provide a suitable rebuttal to their concerns.  

We look forward to receiving your revised manuscript.

Kind regards,

Abhinava Kumar Mishra, PhD

Academic Editor

PLOS ONE

 [This research was supported by the National Natural Science Foundation of China (No. 32000832), the Natural Science Foundation of Shandong Province (No. ZR2020QC096.].  

[This work is supported by the Province Natural Science Foundation of Shandong (No. ZR2020QC096)

and National Natural Science Foundation of China (NSFC) (No. ]

 [This research was supported by the National Natural Science Foundation of China (No. 32000832), the Natural Science Foundation of Shandong Province (No. ZR2020QC096.].

Reviewers' comments:

Reviewer's Responses to Questions

**Comments to the Author**

1. Is the manuscript technically sound, and do the data support the conclusions?

Reviewer #1: Yes

Reviewer #2: Yes

2. Has the statistical analysis been performed appropriately and rigorously? 

Reviewer #1: Yes

Reviewer #2: Yes

3. Have the authors made all data underlying the findings in their manuscript fully available?

Reviewer #1: Yes

Reviewer #2: No

4. Is the manuscript presented in an intelligible fashion and written in standard English?

Reviewer #1: Yes

Reviewer #2: No

5. Review Comments to the Author

Reviewer #1: In this article, authors firstly conformed that the muscle TOR gene played an important role in endurance exercise against HSD-induced age-related skeletal muscle degeneration, as it determined the activity of the MTOR/SIR2/PGC-1α and MTOR/ATG2/PGC-1α pathways in skeletal muscle. The author provides a lot of data to support his conclusion. However, there were still some minor issues that need to be fixed to make the article better.

1. The shortened words should be standardized and uniform. For example, the article had TOR and MTOR, these two have the same meaning or different meaning.

2. The format and grammar of the article need to be carefully revised.

3. The author should add some latest references for discussion and analysis.

Reviewer #2: The manuscript by Wang et al., studies the mechanism how exercise can ameliorate age-related skeletal muscle deterioration by high salt diet. This study can provide novel insight into how sarcopenia is caused. However there are several issues that need to be addressed:

Major issues

1. The reason why the authors used flies where TOR was either knocked-down or overexpressed for the experiments is unclear. The authors need to provide a better explanation. At the same time the authors need to add results of experiments obtained from wild type flies. Also, the authors need to prove that the RNAi for TOR indeed knocked-down the gene.

2. MHC-GAL4 expression in not limited to skeletal muscles. It is expressed in other sites such as cardiac muscles. The authors need to address that in the text. They also need to use a more specific driver (refer to Zappia and Frolov, 2016) and repeat some of their more fundamental experiments.

Minor Issues:

1. The manuscript has several grammatical errors.

2. The authors need to provide the full forms of abbreviations while mentioning them for the first time.

3. The authors need to make sure that they consistently use the same nomenclature to refer to the same thing. For isntance they have both used weeks and weekends to denote age of flies.

4. The authors need to make sure that they used proper terminology throughout the paper consistent with previous publications. Also, they need to represent the genotypes of the flies correctly throughout the manuscript. Especially it will be helpful to have complete genotypes in the figures.

5. Information about rearing such as day-night cycle, temperature and humidity have not been mentioned.

6. PLOS authors have the option to publish the peer review history of their article (what does this mean?). If published, this will include your full peer review and any attached files.

Reviewer #1: No

Reviewer #2: No

---

## [Author Response · Author response to Decision Letter 0]

9 Aug 2024

Reviewer #1: In this article, authors firstly conformed that the muscle TOR gene played an important role in endurance exercise against HSD-induced age-related skeletal muscle degeneration, as it determined the activity of the MTOR/SIR2/PGC-1α and MTOR/ATG2/PGC-1α pathways in skeletal muscle. The author provides a lot of data to support his conclusion. However, there were still some minor issues that need to be fixed to make the article better.

1. The shortened words should be standardized and uniform. For example, the article had TOR and MTOR, these two have the same meaning or different meaning.

The full names of TOR and MTOR have been added to distinguish them

2. The format and grammar of the article need to be carefully revised.

The format has been modified according to the requirements of the journal, and the syntax has been Revised. Please see the blue font in the body of the Revised Manuscript with Track Changes

3. The author should add some latest references for discussion and analysis.

It has been modified. For details, see the section highlighted in blue in Revised Manuscript with Track Changes.

Reviewer #2: The manuscript by Wang et al., studies the mechanism how exercise can ameliorate age-related skeletal muscle deterioration by high salt diet. This study can provide novel insight into how sarcopenia is caused. However there are several issues that need to be addressed:

Major issues

1. The reason why the authors used flies where TOR was either knocked-down or overexpressed for the experiments is unclear. The authors need to provide a better explanation. At the same time the authors need to add results of experiments obtained from wild type flies. Also, the authors need to prove that the RNAi for TOR indeed knocked-down the gene

TOR knockdown/overexpression flies were used to verify the role of TOR in this study. For example, from the differences in motor performance, antioxidant capacity and gene expression levels between the TOR overexpression group and the TOR normal expression group, we can see the role of TOR genes in these aspects..The construction of TOR overexpression/knockdown system was demonstrated by comparing the expression level of MTOR gene in flies with normal TOR expression group at three week-old without intervention.The results showed that there were significant differences in the expression level of MTOR gene between overexpression group/knockdown group and normal expression group(See text 3.1 for details).

2. MHC-GAL4 expression in not limited to skeletal muscles. It is expressed in other sites such as cardiac muscles. The authors need to address that in the text. They also need to use a more specific driver (refer to Zappia and Frolov, 2016) and repeat some of their more fundamental experiments.

The experimental methods and procedures related to skeletal muscle have been completed (related to overexpression/knockdown MTOR gene expression levels in Drosophila melanogaster over three weeks).We do not deny that MHC-Gal4 is expressed in other muscles. However, we only examined the skeletal muscle of fruit flies, and we plan to study the specific changes of TOR gene in other muscles in the future.

Minor Issues:

1. The manuscript has several grammatical errors.

The format has been modified according to the requirements of the journal, and the syntax has been Revised. Please see the blue font in the body of the Revised Manuscript with Track Changes

2. The authors need to provide the full forms of abbreviations while mentioning them for the first time.

It has been modified. For details, see the section highlighted in blue in Revised Manuscript with Track Changes.

3. The authors need to make sure that they consistently use the same nomenclature to refer to the same thing. For isntance they have both used weeks and weekends to denote age of flies.

It has been modified. For details, see the section highlighted in blue in Revised Manuscript with Track 

4. The authors need to make sure that they used proper terminology throughout the paper consistent with previous publications. Also, they need to represent the genotypes of the flies correctly throughout the manuscript. Especially it will be helpful to have complete genotypes in the figures.

It has been modified. For details, see the section highlighted in blue in Revised Manuscript with Track 

5. Information about rearing such as day-night cycle, temperature and humidity have not been mentioned.

For details, see Revised Manuscript with Track Changes 2.1.

The information the editor says about the journal format, funding, supporting documents, etc. has been corrected.

---

## [Decision Letter · Decision Letter 1]

16 Sep 2024

Muscular TOR knockdown and endurance exercise ameliorate high salt and age-related skeletal muscle degradation by activating the MTOR-mediated pathway

PONE-D-24-20830R1

Dear Dr. Wang,

We’re pleased to inform you that your manuscript has been judged scientifically suitable for publication and will be formally accepted for publication once it meets all outstanding technical requirements.

Kind regards,

Abhinava Kumar Mishra, PhD

Academic Editor

PLOS ONE

Additional Editor Comments (optional):

Reviewers' comments:

Reviewer's Responses to Questions

**Comments to the Author**

1. If the authors have adequately addressed your comments raised in a previous round of review and you feel that this manuscript is now acceptable for publication, you may indicate that here to bypass the “Comments to the Author” section, enter your conflict of interest statement in the “Confidential to Editor” section, and submit your "Accept" recommendation.

Reviewer #1: All comments have been addressed

Reviewer #2: All comments have been addressed

2. Is the manuscript technically sound, and do the data support the conclusions?

Reviewer #1: Yes

Reviewer #2: Yes

3. Has the statistical analysis been performed appropriately and rigorously? 

Reviewer #1: Yes

Reviewer #2: Yes

4. Have the authors made all data underlying the findings in their manuscript fully available?

Reviewer #1: Yes

Reviewer #2: No

5. Is the manuscript presented in an intelligible fashion and written in standard English?

Reviewer #1: Yes

Reviewer #2: Yes

6. Review Comments to the Author

Reviewer #1: This study confirms that the muscle TOR gene plays an important role in endurance exercise against HSD-induced age-related skeletal muscle degeneration, as it determines the activity of the mammalian target of rapamycin(MTOR)/SIR2/PGC-1α and MTOR/ATG2/PGC-1α pathways in skeletal muscle. The data supports the conclusions, and the statistical analysis is performed appropriately and rigorously, and the manuscript is presented in an intelligible fashion and written in standard English after the paper has been revised.

Reviewer #2: The authors have satisfactorily addressed all my concerns and have also made the necessary changes. The authors need to share the data sets and provide a summary table for the stats.

7. PLOS authors have the option to publish the peer review history of their article (what does this mean?). If published, this will include your full peer review and any attached files.

Reviewer #1: No

Reviewer #2: No

---

## [Editor Report · Acceptance letter]

24 Sep 2024

PONE-D-24-20830R1 

PLOS ONE

Dear Dr. Wang, 

I'm pleased to inform you that your manuscript has been deemed suitable for publication in PLOS ONE. Congratulations! Your manuscript is now being handed over to our production team.

Kind regards, 

on behalf of

Dr. Abhinava Kumar Mishra 

Academic Editor

PLOS ONE